# Re-organization of Pacific overturning circulation across the Miocene Climate Optimum

Ann Holbourn [1] ✉, Wolfgang Kuhnt [1], Denise K. Kulhanek [1], Gregory Mountain [2], Yair Rosenthal[2,3], Takuya Sagawa[4], Julia Lübbers[1,5] & Nils Andersen [6]

The response of the ocean overturning circulation to global warming remains controversial. Here, we integrate a multiproxy record from International Ocean Discovery Program Site U1490 in the western equatorial Pacific with published data from the Pacific, Southern and Indian Oceans to investigate the evolution of deep water circulation during the Miocene Climate Optimum (MCO) and Middle Miocene Climate Transition (MMCT). We find that the northward export of southern-sourced deep waters was closely tied to high-latitude climate and Antarctic ice cover variations. Global warming during the MCO drove a progressive decrease in carbonate ion concentration and density stratification, shifting the overturning from intermediate to deeper waters. In the western equatorial Pacific, carbonate dissolution was compensated by increased pelagic productivity, resulting in overall elevated carbonate accumulation rates after ~16 Ma. Stepwise global cooling and Antarctic glacial expansion during the MMCT promoted a gradual improvement in carbonate preservation and the initiation of a near-modern Pacific overturning circulation. We infer that changes in the latitudinal thermal gradient and in Southern Ocean zonal wind stress and upper ocean stratification drove radically different modes of deep water formation and overturning across the MCO and MMCT.

The future response of the ocean meridional overturning circulation to recent increases in temperature and atmospheric $p\mathrm{CO_2}$, driving accelerated melting of ice shelves and continental ice, is an issue of intense debate[1–5]. The complexity of such a highly dynamic component of the climate system restricts, in particular, the predictability of critical thresholds that trigger abrupt climate change once they are crossed[6]. Paleoclimate reconstructions have revealed that the Earth's climate and ocean circulation have changed dramatically over the past ~66 million years[7,8]. Even though past climates do not provide direct analogs for present-day conditions, they offer key insights into the sensitivity to changing boundary conditions, such as temperature, ice volume, and greenhouse gas concentrations, and into the scale, duration, and driving mechanisms of change across a wide range of mean background states. Reconstructing past warm climates, therefore, helps us understand the response of the ocean–climate system to atmospheric $p\mathrm{CO_2}$

[1]Institute of Geosciences, Christian-Albrechts-University, D-24118 Kiel, Germany. [2]Department of Earth and Planetary Sciences, Rutgers, The State University of New Jersey, Piscataway, NJ, USA. [3]Department of Marine and Coastal Sciences, Rutgers, The State University of New Jersey, New Brunswick, NJ, USA. [4]Faculty of Geosciences and Civil Engineering, Institute of Science and Engineering, Kanazawa University, Kanazawa, Japan. [5]Center for Marine and Environmental Research (CIMA), University of Algarve, Faro, Portugal. [6]Leibniz Laboratory for Radiometric Dating and Stable Isotope Research, Christian-Albrechts-University Kiel, D-24118 Kiel, Germany. ✉e-mail: ann.holbourn@ifg.uni-kiel.de

concentrations beyond the instrumental range and also provides constraints on future changes.

The Miocene Epoch (23.03–5.33 Ma) was characterized by atmospheric $p$CO$_2$ and global temperatures that were similar to or higher than today's[9,10] and was also marked by major climate reversals associated with fundamental changes in ice volume and ocean circulation[8,11–13]. The interval ~17–13 Ma is of specific interest, as it encompasses the warmest phase of the past 35 million years, the Miocene Climate Optimum (MCO, ~16.9–14.7 Ma), which temporarily reversed the long-term global cooling trend over the Cenozoic Era[8,12–15]. Estimates of global mean surface air temperature during the MCO are at least 10 °C higher than present-day values[10,16]. This prolonged interval of warmth ended with global cooling, coupled with a stepwise increase in Antarctic ice volume during the Middle Miocene Climate Transition (MMCT, ~14.7–13.8 Ma)[8,12–15]. Based on the co-evolution of benthic foraminifer δ$^{18}$O, atmospheric $p$CO$_2$, and global temperature trends with ocean crustal production rates, a recent study suggested that changes in tectonic degassing were the prime driver of climate evolution during the MCO and MMCT[10]. Previous studies also proposed that the main phase of Columbia River basalt magmatism between ~16.9 and 16 Ma[17] contributed to CO$_2$ forcing during the early part of the MCO[17–19]. However, the evolution of the overturning circulation and deep water chemistry across these major climate reversals remains poorly understood due to the scarcity of continuous, well-dated sedimentary records spanning the MCO and MMCT.

As the world's largest ocean, the Pacific Ocean strongly influences the global transport of heat, oxygen, and nutrients and the evolution of the Earth's carbon cycle[20,21]. The modern Pacific overturning circulation is dominated by the influx of deep water masses from the Southern Ocean along the western Pacific margin from New Zealand to the Philippines (Antarctic Bottom Water: AABW, and Circumpolar Deep Water: CDW)[22–26] (Supplementary Note 1). As these deep water masses move northwards, they warm up and rise through diapycnal diffusion, reaching far into the Northern Hemisphere before they become entrained into the North Pacific anticyclonic gyre to eventually contribute to the southward return flow of Pacific Central Water (PCW). Today, ~10 Sv of Pacific thermocline waters originating from

the North Pacific enter the Indian Ocean via the Indonesian Throughflow, forming the principal return flow of the global thermohaline circulation. An inflow of deep water (~2 Sv) from the tropical Pacific Ocean into the Banda Sea additionally occurs through the Lifamatola Passage, south of Sulawesi, at depths below ~1250 m[27]. By contrast, the tropical connection between the Pacific and Indian Oceans remained widely open during the Early to Middle Miocene[28] and, unlike today, a broad, deep inter-ocean connection existed via the Indonesian Gateway, promoting advection of intermediate and deep water masses into the Indian Ocean.

Drilling at Site U1490 (05°48.95′N, 142°39.27′E, 2341 m water depth) during the International Ocean Discovery Program (IODP) Expedition 363 recovered a continuous, carbonate-rich sediment succession from the western equatorial Pacific Ocean, which provides a detailed archive of past changes in ice volume, ocean chemistry and circulation through the Miocene[29]. During the middle Miocene, Site U1490 had a backtracked paleo-latitude of 3–4°N with paleo-longitude and water depth close to that of the present day (Supplementary Note 7). The upper Lower to Middle Miocene interval (~18 to 13 Ma) targeted in this study was drilled with the advanced piston corer (APC) and half-length advanced piston corer (HAPC)[29]. Foraminifer preservation within this interval is moderate to good, with dissolution primarily affecting thin-walled planktic foraminifer tests over parts of the succession, whereas benthic foraminifer tests are generally well-preserved and translucent with minimal overgrowth. Recrystallization, overgrowth, and cementation only become severe below the level of APC/HAPC refusal in the Lower Miocene interval (prior to ~18 Ma). The typical preservation states of benthic foraminifer species selected for isotope analyses are documented in Supplementary Note 3.

Site U1490 is located on the northern edge of the Eauripik Rise in the Caroline Basin north of Papua New Guinea within the direct flow path of southern-sourced deep-water masses into the western equatorial Pacific Ocean[23,24,30] (Fig. 1, Supplementary Note 1). Seismic profiles revealed current-controlled waves in carbonate sediments on the northern Eauripik Rise, which are indicative of a vigorous bottom current flow through the late Early to Middle Miocene[31]. This strategic location makes Site

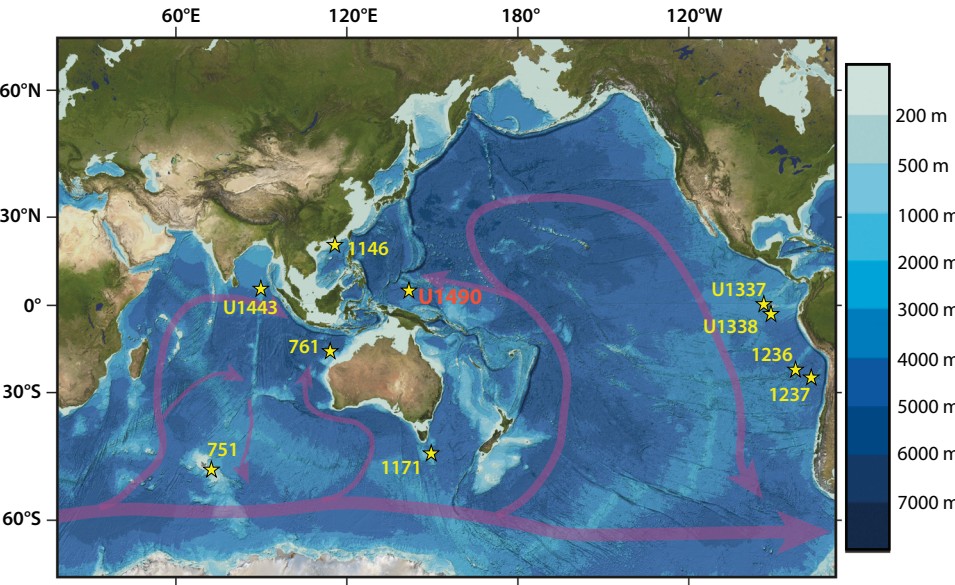

**Fig. 1 | Locations of International Ocean Discovery Program Site U1490 and other sites discussed in this study with simplified Pacific and Indian Ocean deep water circulation paths.** Bathymetry is based on gridded bathymetric data sets from the General Bathymetric Chart of the Oceans (GEBCO Compilation Group, GEBCO 2024 Grid (https://doi.org/10.5285/1c44ce99-0a0d-5f4f-e063-

7086abc0ea0f); land topography is from the Blue Marble satellite mosaic, courtesy of NASA's Earth Observatory (https://neo.gsfc.nasa.gov/view.php?datasetId=BlueMarbleNG-TB), and World Vector Shoreline, National Geophysical Data Center (https://shoreline.noaa.gov/data/datasheets/wvs.html).

U1490 ideally suited to reconstruct the variability of Southern Ocean deep water export and the evolution of the Pacific overturning circulation under reversing climate trends when ice volume, global temperatures, and greenhouse gas concentrations varied markedly[8–15]. In this work, we reconstruct the evolution of western equatorial Pacific deep-water chemistry across the MCO and MMCT, using high-resolution benthic foraminifer stable isotopes, X-ray fluorescence (XRF) scanner-derived elemental records, light reflectance spectroscopy data as well as coarse fraction >63 μm residue and calcium carbonate (CaCO$_3$) weight percentage records from Site U1490. We integrate our results with published records from key locations in the Pacific, Indian, and Southern Oceans to investigate ocean-wide circulation changes and to better understand the processes driving short- and long-term variations in deep water production and ocean overturning on a warmer-than-modern Earth.

## Results and discussion

### Western Pacific Warm Pool deep sea stable isotope reference

Previous studies of long-term climate variability in the Western Pacific Warm Pool (WPWP) primarily focused on Ocean Drilling Program Site 806 (0°19.1′N, 159°21.7′E, 2534 m water depth). Site 806, located ~1050 nm southeast of Site U1490, has served as a warm end-member to monitor the evolution of Neogene zonal and meridional gradients[29,32–35]. However, the lower Middle to Lower Miocene interval at Site 806 is increasingly lithified downcore and cannot be tied to an astronomical target for precise dating due to incomplete recovery. Here, we present astronomically tuned stable isotope reference records for the WPWP region from Site U1490, which span the interval 18.2–12.5 Ma (Fig. 2a, e, Supplementary Note 2, Supplementary Data File 1). The highly resolved benthic foraminifer δ$^{18}$O and δ$^{13}$C time series (~4 kyr resolution over most of the record) from Site U1490 track the onset and development of the MCO (~16.9–14.7 Ma) and the stepwise global cooling and glacial expansion across the MMCT (~14.7–13.8 Ma). As shown in previous studies[8,36], lower mean δ$^{18}$O characterizes the initial warmer phase of the MCO between ~16.9 and 16 Ma, while high-amplitude δ$^{18}$O decreases (hyperthermals) are primarily paced by short eccentricity during the remainder of the MCO (Fig. 2e, Supplementary Notes 2, 3). Stepwise global cooling and glacial expansion during the MMCT culminated with a major δ$^{18}$O increase from ~13.9 to 13.8 Ma. The δ$^{13}$C signal additionally captures the long-lasting Monterey Excursion[36,37] from ~16.7 to 13.5 Ma, which consists of eight distinctive carbon isotope maxima (CM) exhibiting the characteristic imprint of the ~400 kyr eccentricity cycle with a lag of ~20–50 kyr to eccentricity (Fig. 2a, Supplementary Notes 2, 3). Carbon isotope maximum 4b is not, however, fully discernible in our record, as it falls within a disturbed interval in Section 1 of Core U1490B-27F.

The Site U1490 δ$^{18}$O and δ$^{13}$C records provide unequaled resolution across the interval ~17.5 to 15.7 Ma, which is poorly represented in deep-sea sediment cores, as it includes a prolonged interval of intense carbonate dissolution (Lavender unconformity[38,39]). The onset and development of the warmest phase of the MCO (~16.9–16 Ma) as well as the Mi2 δ$^{18}$O maximum[11,40–42], which terminated this initial period of peak warmth at ~16 Ma, are especially well expressed at Site U1490 (Figs. 2, 3), enabling an improved correlation to the astronomical solution[43] across the interval ~17.5 to 15.7 Ma (Supplementary Note 2). The Site U1490 δ$^{18}$O record further reveals that 100 kyr eccentricity-driven climate cycles, encompassing the Mi2 event, exhibit characteristic saw-tooth features (Fig. 4), implying that ice sheet build-up may have been the prime driver of this event rather than a decrease in deep water temperature. By contrast, glacial–interglacial cycles are overall more symmetrical and lack saw-tooth features after 15.9 Ma (Figs. 2, 3), suggesting that transient deep water warming was the main cause of hyperthermal δ$^{18}$O decreases during the remainder of

the MCO. The δ$^{18}$O increase across the Mi2 event (~16 Ma), previously estimated to represent a sea-level fall of ~40 m[11], marked a substantial step in Antarctic ice sheet development. Consistently higher glacial δ$^{18}$O values (by ~0.2‰) following the Mi2 glaciation (Figs. 2, 3) suggest that the ice sheet became larger, more stable, and overall less susceptible to summer radiative forcing after ~16 Ma than during the initial, warmer phase of the MCO (~16.9–16.1 Ma).

The Site U1490 record additionally offers the opportunity to directly correlate a well-resolved, astronomically tuned isotope stratigraphy to the global magnetic polarity timescale[44,45] over the late Early to Middle Miocene interval 17–13 Ma (Supplementary Note 2). The ages of polarity chrons between 17 and 15 Ma are still under debate, as they have been calibrated through the correlation of seafloor magnetic anomalies to a few radio-isotopic ages, assuming constant spreading rates over extended time intervals[44,45]. The interval 17–15 Ma has remained especially problematic for integrating magnetic polarity datums to an astronomically tuned chronology due to the scarcity of continuous sediment successions and the lack of well-resolved magnetostratigraphic and isotopic records. The integration of the Site U1490 shipboard magnetostratigraphy[29] and our isotope cyclostratigraphy, thus, provides additional constraints on the ages of polarity chrons over this enigmatic period (Supplementary Note 2). The development of an integrated cyclostratigraphic and magnetostratigraphic timescale is essential for the accurate correlation of climatic events on a global scale.

### Evolution of western Pacific deep water chemistry across the MCO

Changes in deep water properties at Site U1490 signal the emergence of new circulation patterns in the western equatorial Pacific Ocean during the MCO. Our high-latitude climate and deep water chemistry proxy records suggest that fluctuations of the Antarctic ice sheet and changes in the latitudinal thermal gradient exerted a major control on the mode of deep water formation and the strength of Pacific overturning. During the initial, warmer phase of the MCO (~16.9–16 Ma), our combined benthic foraminifer δ$^{18}$O, Log(Si/Ti), carbonate fraction >63 μm residue, and CaCO$_3$ weight percentage records show a distinctive response to 100 kyr eccentricity and 41 kyr obliquity forcing (Figs. 2, 3, Supplementary Notes 3–6). Between ~16.9 and 16 Ma, warmer intervals are characterized by increased concentration of biogenic silica and improved carbonate preservation of planktic foraminifer assemblages at eccentricity/obliquity maxima (Figs. 2, 3). Conversely, silica accumulation decreased, and carbonate dissolution increased during colder intervals at eccentricity/obliquity minima, as shown by an increased abundance of etched or fragmented planktic foraminifers.

Following the Mi2 glacial expansion at ~16 Ma, the Site U1490 Log(Si/Ti) and carbonate fraction >63 μm weight percentages exhibit muted responses to eccentricity and obliquity forcing (Figs. 2, 3). Carbonate fraction >63 μm weight percentages decrease, remaining mostly between ~5% and 1% from ~16 to 14.6 Ma, while carbonate concentration increases from 90% to 95%, and carbonate accumulation rates also show a distinct rise from ~16 to 14.8 Ma (Figs. 2, 3, Supplementary Fig. S5-1d). Coarse fraction residues contain abundant radiolarians and predominantly etched and fragmented planktic foraminifers within this interval. Planktic foraminifer assemblages are dominated by the large tests of robust species such as *Dentoglobigerina altispira*, *Dentoglobigerina venezuela* and *Sphaeroidinellopsis* spp. that live deeper in the water column, are generally more resistant to carbonate dissolution and may also be indicative of high productivity conditions[29]. These features indicate that a more corrosive, carbonate ion undersaturated deep water mass prevailed in the western equatorial Pacific Ocean after ~16 Ma and through the remainder of the MCO. A

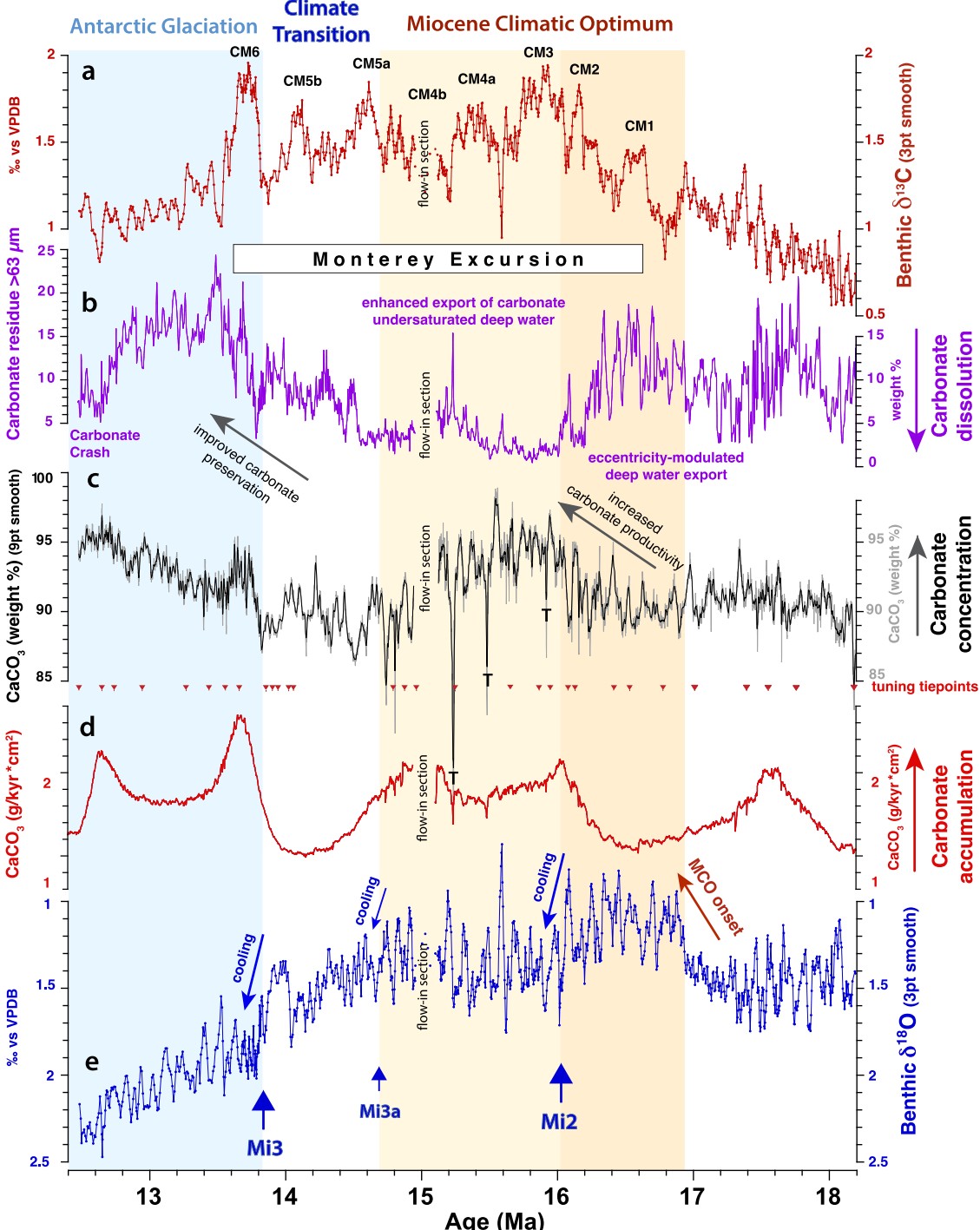

**Fig. 2 | Evolution of deep water chemistry at West Pacific Warm Pool Site U1490 over the interval ~18.2–12.5 Ma. a** Benthic foraminifer δ¹³C (Supplementary Data File 1); 3 pt smooth = 3 point moving average. Monterey Excursion carbon isotope maxima[82,91] are labeled CM1, 2, 3, 4a, 4b, 5a, 5b, and 6. **b** Fraction >63 μm of carbonate (weight%) as indicator for carbonate dissolution (Supplementary Data File 1). **c** Carbonate concentration (weight%) based on X-ray fluorescence scanner-derived Log(Ca/Al+Si) calibrated with carbonate concentration measurements of discrete samples (Supplementary Data Files 4, 5); 9 pt smooth = 9 point moving average. **d** Carbonate accumulation rates calculated with sedimentation rates based on a smooth curve fit through the tuning tie points (Supplementary Note 5) and dry bulk density based on shipboard gamma ray attenuation measurements[29]. **e** Benthic foraminifer δ¹⁸O (Supplementary Data File 1); 3 pt smooth = 3 point moving average. Blue arrows mark δ¹⁸O increases associated with Antarctic ice sheet expansion (Mi2, Mi3a, and Mi3 glaciation events following refs. 11,40–42). Warmer phase of Miocene Climate Optimum (MCO) from ~16.9 to 16 Ma shaded dark orange; hyperthermal phase of MCO following Mi2 glaciation shaded light orange. Expansion of Antarctic ice sheet following Mi3 shaded light blue. T: volcanic tephra layer.

similar decrease in the weight of coarse fraction >63 μm weight percentages between ~16 and 14.7 Ma at eastern Indian Ocean Site 761[46], located at comparable paleodepth, suggests that this corrosive deep water expanded throughout the Indo-Pacific region. The

deeper, wider connection between the Pacific and Indian Oceans via the Indonesian Gateway would have enhanced the inter-ocean transfer of southern-sourced deep water during the MCO. Alternatively, Site 761 may have been more influenced by the direct

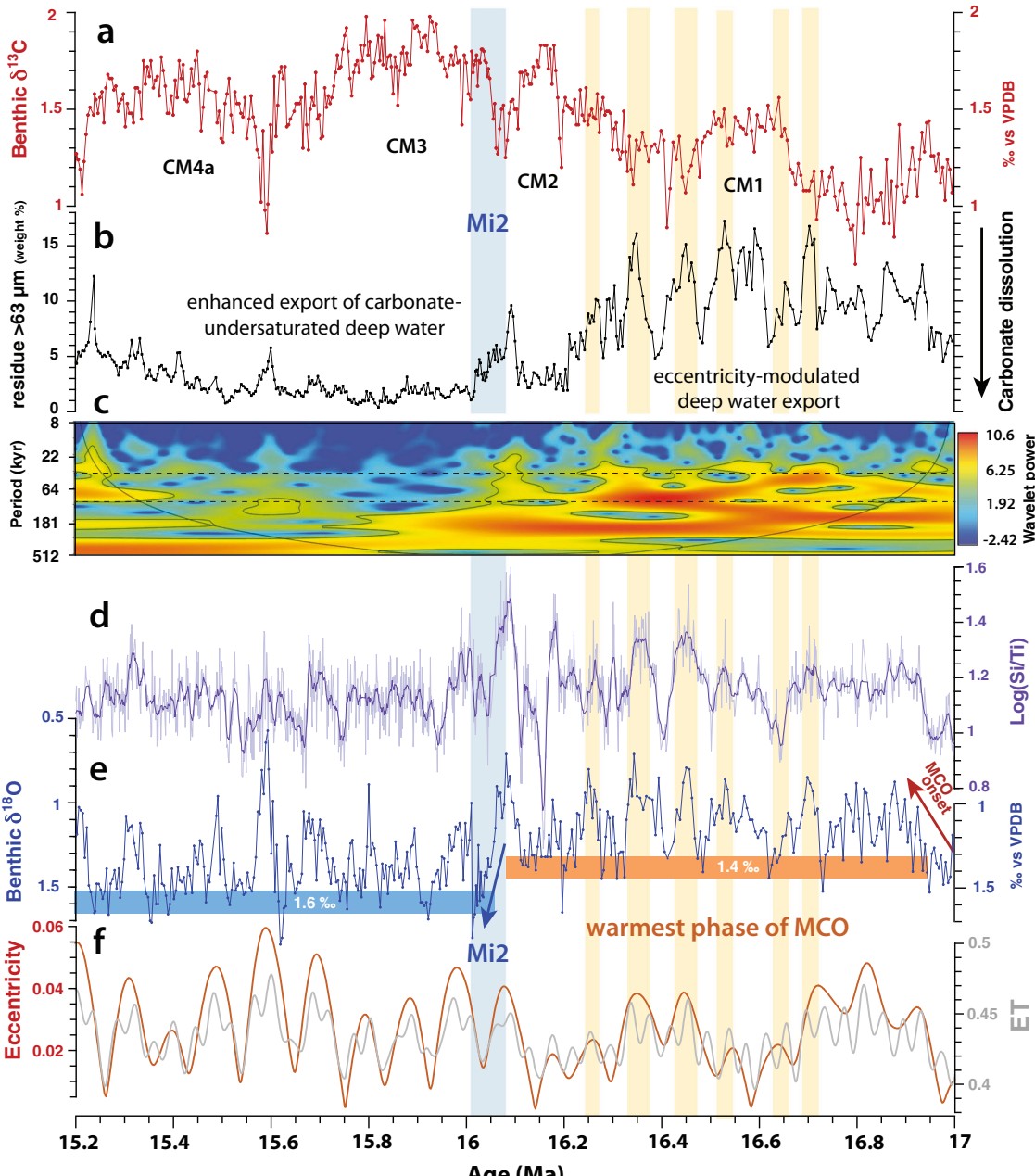

**Fig. 3 | Expanded view of interval 17–15.2 Ma at West Pacific Warm Pool Site U1490. a** Benthic foraminifer δ13C (Supplementary Data File 1). Monterey Excursion carbon isotope maxima[82,91] are labeled CM1, 2, 3, 4a. **b** Weight percentage of >63 µm coarse fraction residues (exclusively composed of complete or fragmented carbonate foraminifer tests) as an indicator for carbonate dissolution (Supplementary Data File 1). **c** Wavelet power of >63 µm coarse fraction residues. Note dominance of 41 kyr (obliquity) and 100 kyr (eccentricity) periodicities (indicated by dashed black lines) and change in amplitude variability across Mi2 glaciation event[11,40–42]. **d** Raw and 9-point moving average X-ray fluorescence scanner-derived Log(Si/Ti) as an indicator of biogenic silicate (opal) (Supplementary Data File 5). **e** Benthic foraminifer δ18O (Supplementary Data File 1). Note δ18O maximum (indicated by blue arrow) and ~0.2‰ increase in δ18O mean following Mi2 glaciation event[11,40–42] (shaded light blue). **f** Eccentricity and eccentricity-obliquity (tilt) composite (ET) from La04[43].

inflow of southern-sourced deep water into the eastern Indian Ocean, as it is today (Fig. 1).

The rise in calcium carbonate concentrations and accumulation rates between ~16 and 15 Ma at Site U1490 (Fig. 2, Supplementary Fig. S5-1d) implies that enhanced dissolution through the water column was compensated by an increase in carbonate primary production in the western equatorial Pacific Ocean. This is corroborated by the composition of planktic foraminifer assemblages and the presence of a large siliceous component in the >63 µm size fractions of sample residues. The paradox of increasing carbonate accumulation rates in a more corrosive Miocene deep ocean environment has been previously explained by a larger increase in carbonate primary production relative to the increase in chemical weathering in a high atmospheric $p$CO2 world[47–49]. Comparison of data- and model-based indicators of global carbon cycle evolution across intervals of major atmospheric CO2 increase and ocean acidification in Earth's history additionally suggested that calcium carbonate ion saturation (Ω) was decoupled from surface ocean pH[49]. While increases of atmospheric $p$CO2 at orbital timescales (20–100 kyr) resulted in a significant pH decrease, Ω remained almost constant at timescales of 20 kyr or even increased at timescales of 40–100 kyr, due to long-term climate and silicate weathering feedbacks[50]. Thus, increased Ω enhanced the productivity

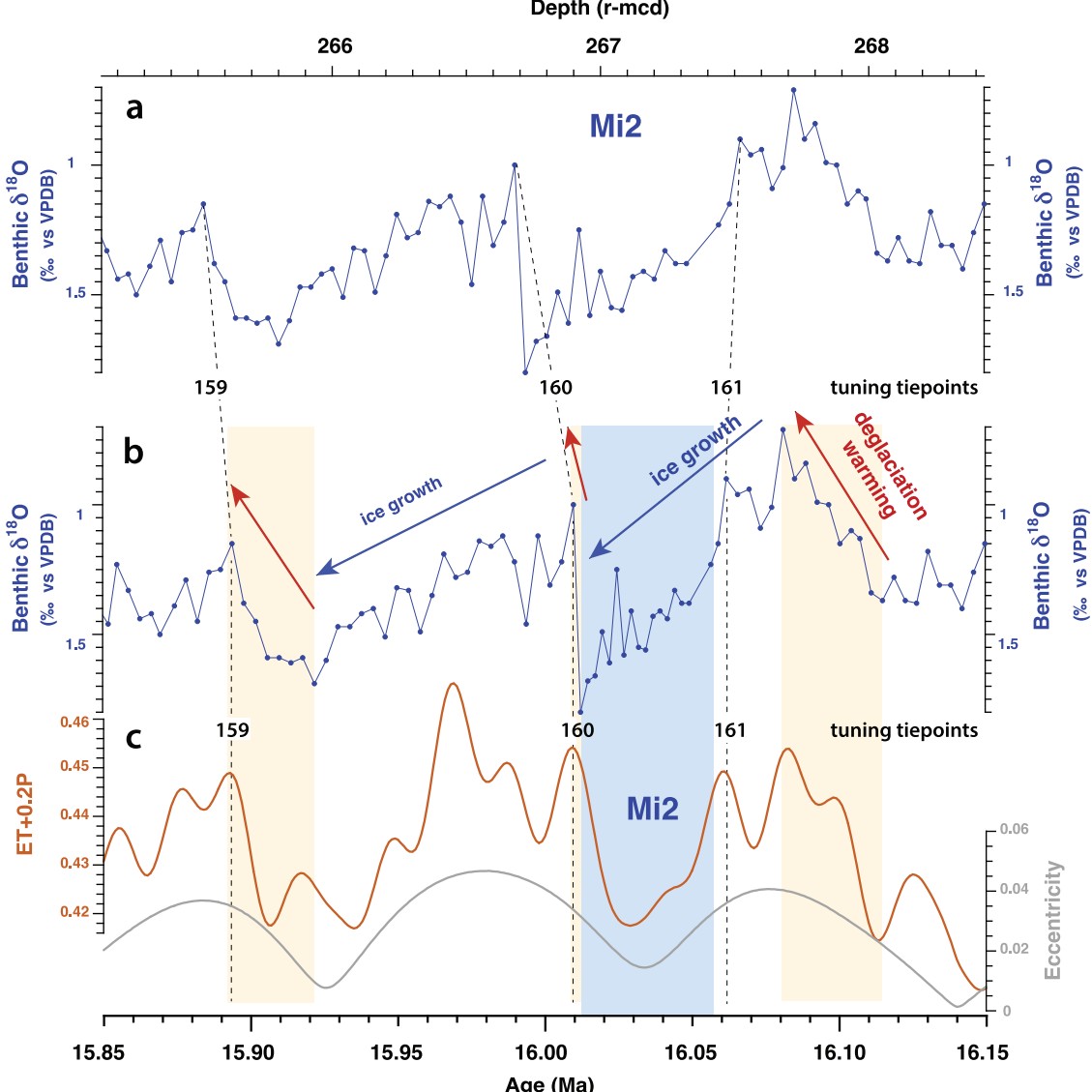

**Fig. 4 | Evolution of Mi2 glaciation event[11,40–42] in relation to La04 orbital parameters[43].** **a** Site U1490 benthic foraminifer $\delta^{18}O$ plotted against depth (revised meter composite depth: r-mcd). **b** Site U1490 benthic foraminifer $\delta^{18}O$ plotted against age (Supplementary Data File 1). **c** Eccentricity (*E*), obliquity (*T*) in radians, and precession parameter P from La04[43] used as tuning target (ET + 0.2P). Glacial terminations are shaded orange and Mi2 glaciation event[11,40–42] shaded light blue. Saw-tooth features in $\delta^{18}O$ between 16.05 and 15.90 Ma suggest the expansion of the Antarctic ice sheet during the Mi2 glaciation event[11,40–42]. Dotted lines mark tuning tie points.

of pelagic calcifying phytoplankton on longer timescales, despite the lower pH. The development of more efficient inorganic carbon concentrating mechanisms to support photosynthetic carbon fixation during periods of high atmospheric $p\mathrm{CO}_2$[51–54] may have additionally contributed to the blooming of calcifying marine phytoplankton.

### Northward expansion of southern-sourced deep-water following Mi2 glaciation

Comparison of astronomically-tuned benthic foraminifer $\delta^{13}C$ records from key locations in the Pacific, Indian, and Southern Oceans with the Site U1490 record provides insight into ocean-wide changes in water mass properties through the MCO (Fig. 5, Supplementary Note 7). The similarity between the $\delta^{13}C$ records from Southern Ocean Sites 751 and 1171 (paleolatitude: ~55–56°S, paleodepth: ~1500 m) and western equatorial Pacific Site U1490 through the MCO (Supplementary Fig. S7-3f-g) demonstrates that Site U1490 remained bathed by southern-sourced waters. The weak $\delta^{13}C$ gradients between Site U1490, eastern Indian Ocean Site 761 (paleolatitude: ~25°S, paleodepth: ~2000 m),

southeastern Pacific Site 1237 (paleolatitude: ~20°S, paleodepth: ~2500 m) and the deeper equatorial Sites U1338, U1443 between ~16 and 15.1 Ma (Fig. 5, Supplementary Fig. S7-3) further support northward expansion of a southern-sourced water mass that filled the Southern Hemisphere and equatorial sectors of the Indian and Pacific Oceans between ~1500 and 4000 m. The convergence of $\delta^{13}C$ gradients between eastern equatorial Pacific Sites U1337–U1338 (paleodepth: >3500 m), equatorial Indian Ocean Site U1443 (paleodepth: >2500 m), and western equatorial Pacific Site U1490 (paleodepth: >2000 m) after ~16 Ma (Fig. 5b, c, g, h, Supplementary Fig. S7-3b-c) also argues for the prevalence of a relatively uniform water mass across the water column at equatorial locations in the Pacific and Indian Oceans until ~15.1 Ma.

By contrast, Site 1146 (paleolatitude: ~23°N, paleodepth: ~2000 m) displays consistently lower $\delta^{13}C$ and $\delta^{18}O$ throughout the MCO (Fig. 5d, e; Supplementary Figs. S7-2a and S7-3a), indicating the presence of a substantially warmer/fresher, nutrient enriched water mass in the northwestern tropical Pacific Ocean, likely originating from

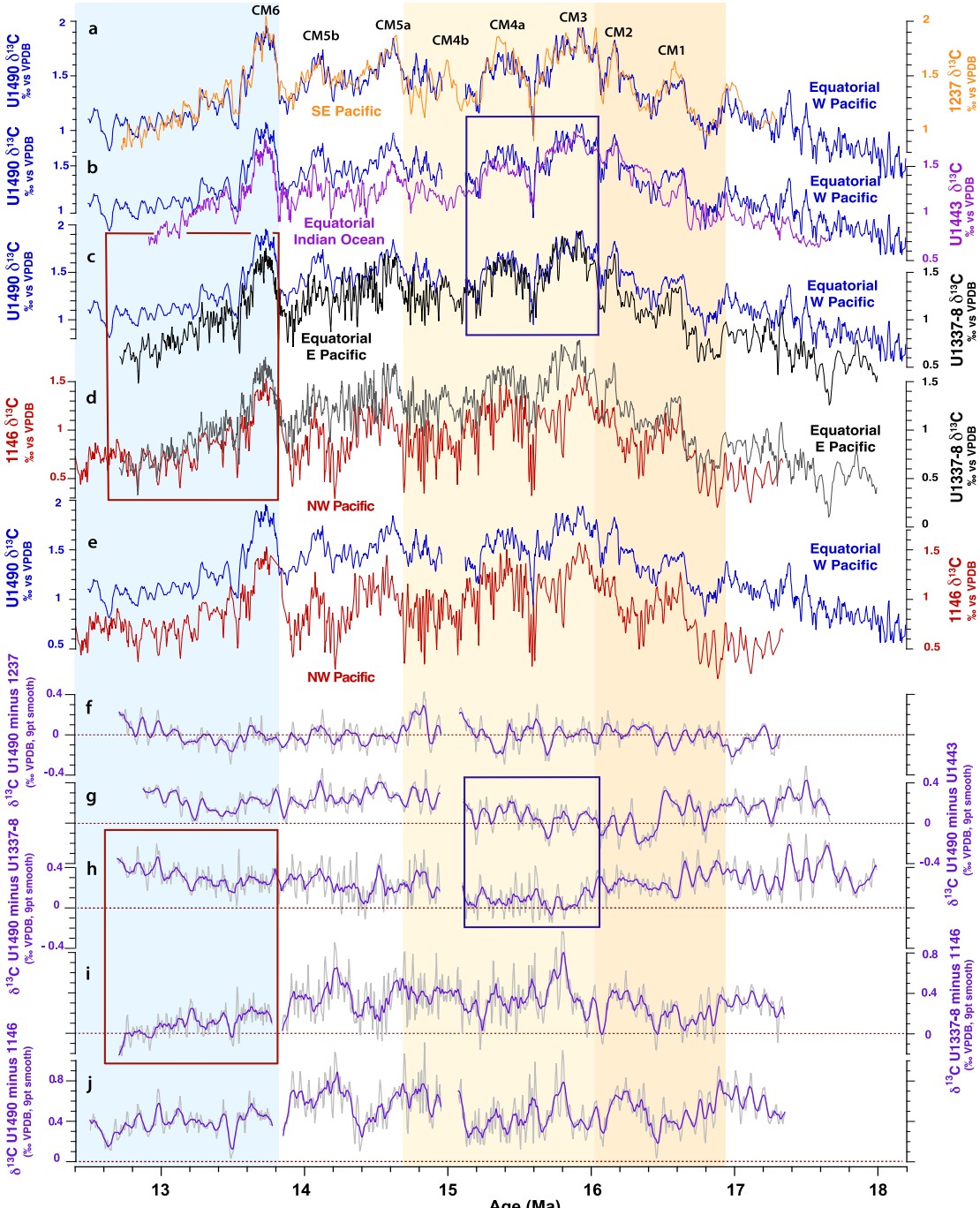

**Fig. 5 | Evolution of Indo-Pacific deep water δ¹³C gradients across the Miocene Climate Optimum (MCO) and Middle Miocene Climate Transition (MMCT).** Comparison of benthic foraminifer δ¹³C records (3 point moving average) from **a** Sites U1490 and 1237[36,76,79,82], **b** Sites U1490 and U1443[36,79,92], **c** Sites U1490 and U1337–U1338[36,71,79,93], **d** Sites 1146[36,79,82] and U1337–U1338[36,71,79,93], **e** Sites U1490 and 1146[36,79,82]. Benthic foraminifer δ¹³C gradients (9 point moving average) between **f** Sites U1490 and 1237, **g** Sites U1490 and U1443, **h** Sites U1490 and U1337–U1338, **i** Sites U1337–U1338 and 1146, **j** Sites U1490 and 1146. Zero line in **f**–**j** is dotted red. Note: convergence of equatorial Pacific and Indian Ocean records between ~16 and 15 Ma (dark blue box) indicates northward expansion of a relatively uniform southern-sourced water mass; convergence of southeastern Pacific Site 1237 and western equatorial Pacific Site U1490 records throughout the MCO and MMCT indicates sustained influence of southern-sourced waters at these locations. Red box marks the increasing advection of δ¹³C depleted water masses into the eastern equatorial Pacific Ocean and the initiation of a near-modern overturning Pacific circulation after -13.8 Ma. Monterey Excursion carbon isotope maxima[82,91] are labeled CM1, 2, 3, 4a, 4b, 5a, 5b, and 6. Warmer phase of MCO from -16.9 to 16 Ma shaded dark orange; hyperthermal phase of MCO following Mi2 glaciation event[11,40–42] shaded light orange. Expansion of Antarctic ice sheet following Mi3 glaciation event[11,40–42] shaded light blue. Stable isotope data on revised age models for Sites 1237, U1443, U1337–U1388 and 1146 are provided in Supplementary Data Files 6, 7, 12, 13.

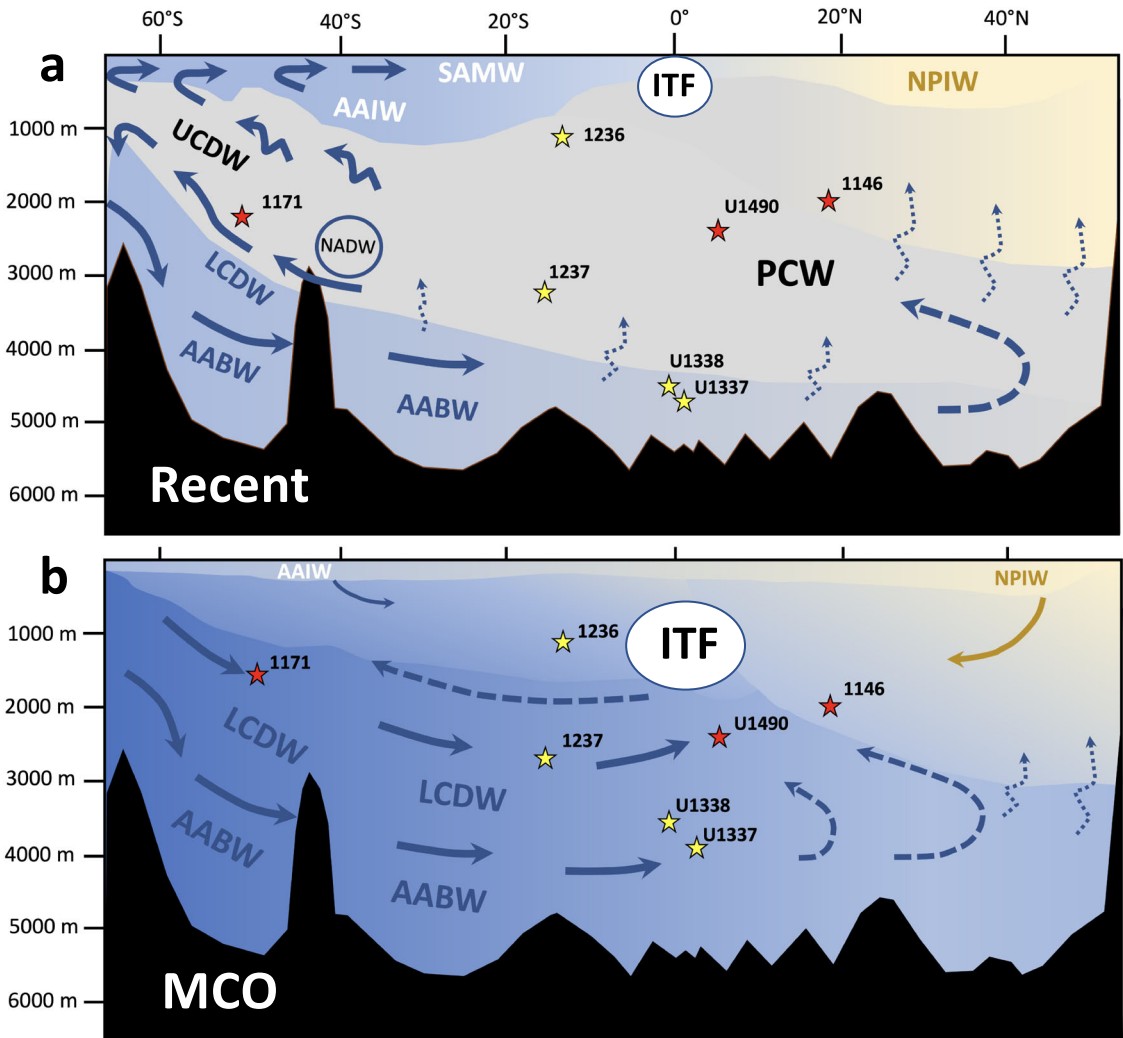

**Fig. 6 | Schematic of Pacific overturning circulation. a** Simplified Pacific circulation today and **b** hypothetical Pacific circulation during the Miocene Climate Optimum (MCO). Main transport directions are indicated by blue arrows. ITF Indonesian Throughflow, which today almost exclusively transports surface and thermocline waters[24]. In the Early to Middle Miocene, the ITF likely included intermediate and deep water masses, which contributed to the global return flow to the Southern Ocean. AABW Antarctic Bottom Water, LCDW and UCDW Lower and Upper Circumpolar Deep Water including laterally advected North Atlantic Deep Water (NADW) today, AAIW Antarctic Intermediate Water, SAMW Subantarctic Mode Water, NPIW North Pacific Intermediate Water, PCW Pacific Central Water, resulting from diffusive mixing of southern-sourced deep water masses with North Pacific nutrient-rich deep and intermediate water masses[23,24,26]. Western Pacific sites are indicated by red stars, eastern Pacific sites by yellow stars. Miocene paleo-depths are from Supplementary Table S7-1.

higher northern latitudes. However, this northern-sourced water mass appears to have remained confined to the Northern Hemisphere during the MCO, implying a different mode of Pacific overturning compared to the present day. Today, cooler, denser Antarctic Bottom Water (AABW) rises through diapycnal diffusion in the tropics and subtropics, as it flows northwards, forming low-density PCW, which eventually returns to the Southern Ocean as part of the Upper Circumpolar Deep Water flow[24] (Fig. 6, Supplementary Note 1). Modeling simulations and proxy data previously suggested that warmer-than-modern climate conditions during the Pliocene promoted the establishment of a large-scale Pacific meridional overturning circulation cell, propelled by convection and deep-water formation in the subarctic Pacific Ocean[55]. Comparison of Pacific $\delta^{13}C$ gradients (Fig. 5, Supplementary Fig. S7-3) indicates, however, that the equatorial and southern Pacific Ocean remained primarily influenced by southern-sourced deep waters during the MCO (Fig. 6) and do not support development of an active cross-equatorial Pacific overturning circulation powered by high-latitude deep convection in the Northern Hemisphere (Fig. 6). Formation of North Pacific deep water may require a cooler Northern Hemisphere with sea ice in the Arctic Ocean and small ice caps over

Greenland and Alaska, which were present during the Pliocene, but did not exist during the warmer MCO[56].

## Mechanisms driving deep water formation and overturning during the MCO

The modern Southern Ocean plays a key role in controlling $CO_2$ storage and regulating whole ocean alkalinity, although the mechanisms driving variability remain poorly understood[26,57,58]. Present-day global warming is projected to drive accelerated melting of ice shelves and enhanced freshwater input to sites of deep-water formation around Antarctica, leading to increased density stratification and a slowdown in ocean overturning and deep water ventilation[3,5,59,60]. However, climate boundary conditions differed markedly during the MCO, when global temperatures and atmospheric $p\mathrm{CO}_2$ were higher, the latitudinal thermal gradient was greatly reduced and the smaller Antarctic ice sheet lacked extensive perennial ice shelves, implying different controls on water mass formation and ocean overturning. In the absence of extended ice shelves during the MCO, freshwater forcing of upper ocean stratification close to Antarctica would have been restricted to relatively

small areas influenced by continental runoff. As a result, density gradients through the water column would have generally decreased, facilitating vertical mixing and deep ocean warming (Fig. 6). This is in sharp contrast to the present-day formation of cold AABW from dense shelf water around the coast of Antarctica[24,61]. Today, AAWB migrates northward into the Pacific Ocean and rises through diapycnal diffusion, as it warms up, leading to the formation of the distinct low-density PCW[24] (Fig. 6). During the MCO, warmer temperatures and decreased stratification in the Southern Ocean would have led to the formation and northward expansion of more homogenous, warmer water masses into the Pacific Ocean (Fig. 6).

The lower latitudinal temperature gradient during the MCO would also have weakened the Southern Hemisphere westerlies that drive the Antarctic Circumpolar Current and the intensity of Southern Ocean upwelling[1,26,62,63], thus affecting intermediate- and deep-water production. A plausible scenario is that the overturning of the Southern Ocean upper cell, which today exports Antarctic Intermediate Water and Subantarctic Mode Water to the lower latitudes[26], became more sluggish during the MCO due to weaker westerlies (Fig. 6). These findings are consistent with recent model simulations, which show that the lower cell (below 2000 m) of the meridional overturning circulation was intensified during the MCO in comparison to pre-industrial, with a distinct maximum centered close to ~30°S[64]. Southward migration of the westerlies and decreased Southern Ocean wind-driven upwelling during the MCO would have also impacted the production and export of deep water from the Polar Antarctic Zone (proto-AABW). Our results indicate that different mechanisms operated during cold and warm climate states from ~16.9 to 16 Ma. The preferential occurrence of carbonate-depleted intervals during colder periods between ~16.9 and 16 Ma (Fig. 3) suggests that the formation of a carbonate undersaturated water mass intensified, when Antarctica retained a sizeable ice cover and sea surface temperatures decreased. By contrast, when global temperatures rose, the latitudinal temperature gradient decreased, the westerlies weakened, Antarctica became almost ice-free and deep water production declined during periods of peak warmth between ~16.9 and 16 Ma (Fig. 3). This scenario is consistent with an abyssal carbonate record from southeastern Indian Ocean Site U1514, which shows that carbonate dissolution occurred preferentially during colder periods at long eccentricity minima between ~17 and 16 Ma[65]. Changes in carbonate content at this location close to the Subantarctic Front were attributed to contraction or expansion of undersaturated deep waters in response to Antarctic ice sheet fluctuations[65].

The Site U1490 records further suggest enhanced formation of carbonate undersaturated deep waters close to Antarctica (proto–AABW) following the Mi2 glacial event at ~16 Ma. The Mi2 ice sheet expansion was coincident with the waning of intense flood basalt volcanism along the Pacific Northwest[17,19], implying a threshold sensitivity of the Antarctic ice sheet to atmospheric $p\text{CO}_2$ and temperature forcing (Fig. 7). We speculate that the larger, more stable ice sheet induced surface cooling close to Antarctica after ~16 Ma, promoting increased formation and northward expansion of corrosive deep waters. The convergence of $\delta^{13}\text{C}$ inter-ocean gradients after ~16 Ma further support that a decrease in Southern Ocean stratification led to the formation and northward expansion of a relatively homogenous deep-water mass into shallower depths in the Pacific and Indian Oceans (Figs. 5, 6, Supplementary Fig. S7-3). This scenario agrees with proxy reconstructions and modeling simulations, which indicated that the Southern Ocean became weakly stratified and better ventilated during periods of warmer climate such as the mid-Pliocene warm period and the MCO[66–68].

Foraminifer-bound nitrogen isotopes from the Pacific, Atlantic and Southern Oceans recently provided evidence that global warming during Miocene and Early Eocene climatic optima led to a contraction of low-latitude oxygen minimum zones[69]. This was attributed to either a decrease in upwelling-driven productivity or an invigoration of deep water ventilation, consistent with weakening of vertical density gradients in the Southern Ocean[69]. Oxygenation and productivity reconstructions, based on planktic foraminifer I/Ca and $\delta^{13}\text{C}$ data[70] and X-ray fluorescence scanner-derived estimates of biogenic silica and carbonate accumulation[71,72] further indicated that a decline in upwelling-driven biological productivity occurred in the eastern tropical Pacific Ocean during the MCO. Our results suggest that the reported changes in ocean oxygenation[69] and upwelling-driven productivity[70] occurred in the context of a wider re-organization of ocean circulation during the MCO. Enhanced formation of deep waters in the Southern Ocean would have promoted a decrease in density stratification, thus shifting the overturning from intermediate to deeper waters in the Pacific and Indian Oceans. This in turn would have resulted in a less dynamic upper ocean overturning, leading to a decrease in upwelling-driven productivity in the eastern tropical Pacific Ocean and improved oxygenation in the deep ocean.

## Towards a modern Indo-Pacific overturning circulation

Between ~14.6 and 13.0 Ma, the Site U1490 Log(Si/Ti), coarse fraction >63 µm residue and carbonate weight percentage records track a progressive decline in biogenic silica accumulation that occurred in tandem with a gradual improvement in carbonate preservation (Figs. 2, 7) and a slight increase in clay content[29]. Planktic foraminifer assemblages within this interval are relatively diverse and better-preserved, including tests of smaller, fragile, dissolution-prone species such as *Trilobatus trilobus, Globigerinoides subquadratus* and *Globigerinoides obliquus*. This assemblage composition indicates an improvement in carbonate preservation, which we relate to an increase in deep water alkalinity after ~14.6 Ma. After ~13 Ma, silica accumulation continued to decrease, whereas carbonate dissolution increased, signaling the onset of a prolonged, global episode of reduced carbonate deposition in the tropical ocean, which has been termed the Carbonate Crash[73]. The progressive decline in biogenic silica accumulation and improvement in carbonate preservation after ~14.6 Ma at Site U1490 occurred in parallel to decreases in global ocean crustal production and tectonic degassing[10] (Fig. 7). We speculate that global cooling and declining atmospheric $p\text{CO}_2$ during the MMCT led to changes in upper ocean stratification and reduced $\text{CO}_2$ uptake at sites of deep water formation in the Southern Ocean, thus inducing a gradual increase in the carbonate ion concentration of deep water. Observations and data-model assimilations have shown that the modern Southern Ocean is highly sensitive to changes in atmospheric $p\text{CO}_2$ and plays a key role in regulating oceanic $\text{CO}_2$ uptake and carbon storage[58,74,75].

Intensification of the latitudinal temperature gradient and of Southern Hemisphere westerlies during the MMCT would have strengthened Southern Ocean upwelling and promoted export of intermediate and deep waters into the equatorial and the northwestern Pacific Ocean, as suggested by previous $\delta^{13}\text{C}$ and neodymium isotope studies[76–79]. The increasing divergence in $\delta^{13}\text{C}$ between Site 1236 intermediate waters and Site 1237 deep waters after ~13.6 Ma (Supplementary Fig. S7-3e) signals advection of Antarctic Intermediate Water at Site 1236 and enhanced upper ocean stratification. Increased vertical partitioning of water masses in the Pacific and Indian Oceans would have stimulated deep ocean carbon storage, further decreasing atmospheric $p\text{CO}_2$ in a positive feedback loop.

Comparison of $\delta^{18}\text{O}$ and $\delta^{13}\text{C}$ profiles between ~15 and 13 Ma further supports that stepwise global cooling and expansion of the Antarctic ice sheet during the MMCT were coupled to major changes in ocean circulation (Fig. 5). Towards the end of the MCO after ~15 Ma, $\delta^{13}\text{C}$ becomes more depleted at eastern equatorial Pacific Sites U1337–U1338 than at Site U1490 (Fig. 5c, h). These trends

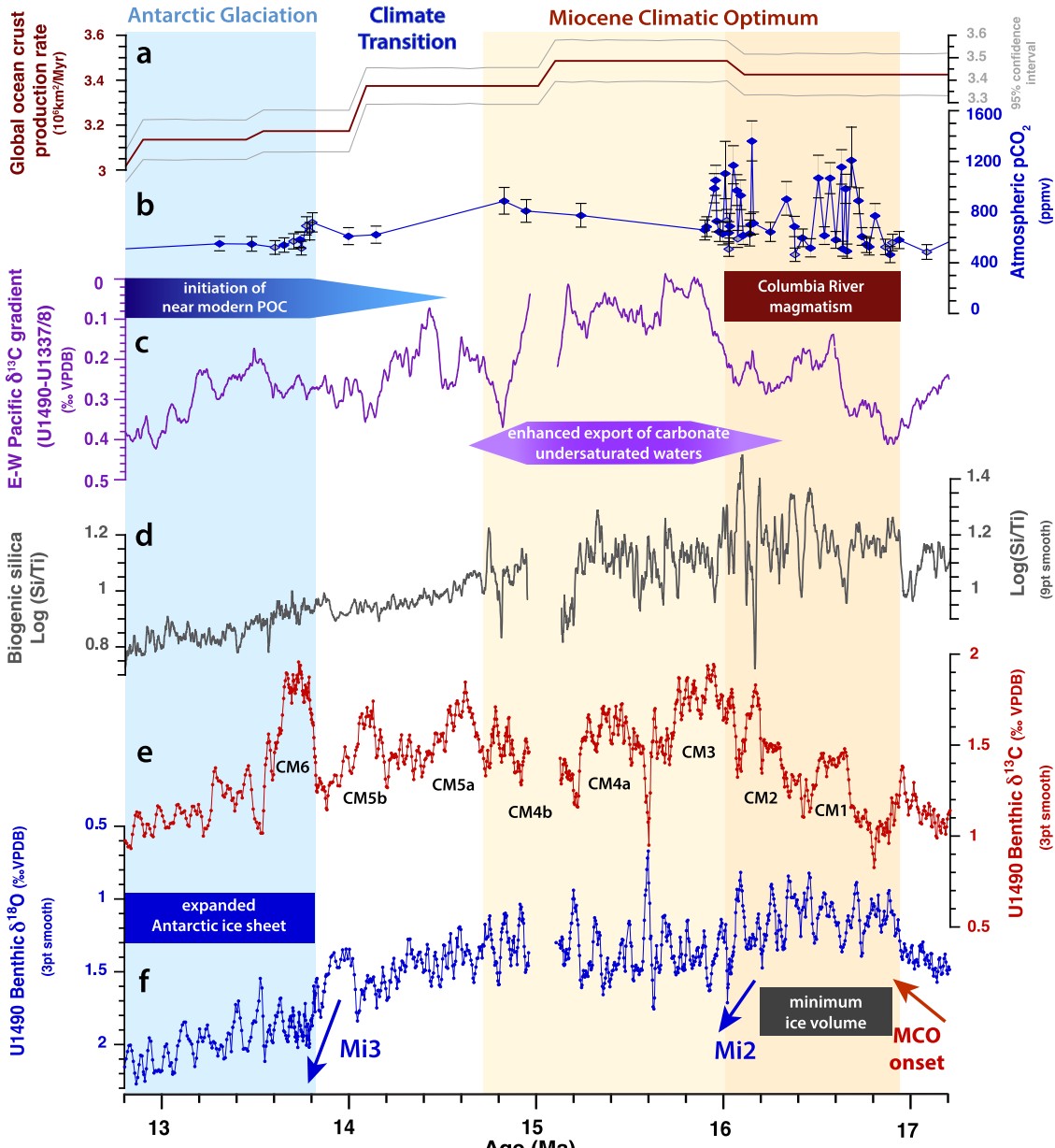

**Fig. 7 | Re-organization of Pacific overturning circulation across the Miocene Climate Optimum (MCO) and Middle Miocene Climate Transition (MMCT).**
**a** Global ocean crust production rate with 95% confidence interval[10]. **b** Boron isotope based atmospheric CO$_2$ estimates across the MCO, calculated from $p$H estimates with constraints from alkalinity, carbonate saturation and the carbonate compensation depth[9]; error bars mark interval between 14th and 86th percentile[9]. Blue diamonds indicate Site 761 data on updated age model (Supplementary Note 2, Supplementary Data File 9). Note: high $p$CO$_2$ variability during the early part of the MCO cannot be directly tied to high-amplitude climate cycles at Site 761, due to the low sedimentation rates (mainly <0.5 cm/kyr) and consequent low resolution of the δ$^{18}$O record (Supplementary Fig. S7-2d). **c** Benthic foraminifer δ$^{13}$C gradient between Site U1490 (western equatorial Pacific) and Sites U1337/U1338 (eastern equatorial Pacific). δ$^{13}$C gradient was calculated in 1 kyr steps using piecewise linear interpolation and a 100 kyr moving average. **d** Site U1490 X-ray fluorescence scanner-derived Log(Si/Ti) as proxy for biogenic silica; 9 pt smooth = 9 point moving average. **e** Site U1490 benthic foraminifer δ$^{13}$C; 3 pt smooth = 3 point moving average. Monterey Excursion carbon isotope maxima[82,91] are labeled CM1, 2, 3, 4a, 4b, 5a, 5b, and 6. **f** Site U1490 benthic foraminifer δ$^{18}$O; 3 pt smooth = 3 point moving average. Note Columbia River magmatism coincident with warmer phase of MCO; waning of volcanism contemporaneous with Mi2 glacial expansion at ~16 Ma suggests threshold response to declining atmospheric $p$CO$_2$. Warmer phase of MCO from ~16.9 to 16 Ma shaded dark orange; hyperthermal phase of MCO following Mi2 glaciation[11,40–42] shaded light orange. Expansion of Antarctic ice sheet following Mi3 glaciation event[11,40–42] shaded light blue. Blue arrows mark δ$^{18}$O increases associated with Antarctic ice sheet expansion (Mi2, Mi3a, and Mi3 following refs. 11,40–42). POC Pacific overturning circulation; ppmv part per million by volume.

become more pronounced following the prominent δ$^{18}$O increase at ~13.8 Ma, corresponding to global cooling and major expansion of the Antarctic ice sheet (Mi3). The δ$^{13}$C offset between Sites U1337–U1338 and Site 1146 gradually disappears after ~13.8 Ma (Fig. 5d, i), indicating the presence of a nutrient-enriched water mass in both the northwestern tropical and the eastern equatorial Pacific Ocean. We attribute these changes to the expansion of a central water mass resembling the modern PCW. Increased advection of δ$^{13}$C depleted water masses into the eastern equatorial Pacific Ocean points to the initiation of a near-modern overturning circulation following Middle Miocene glacial expansion and global cooling.

### Perspective

Our results support that variations in atmospheric $pCO_2$, global temperatures and ice volume strongly influenced Pacific and Indian Ocean overturning pathways on a warmer Miocene Earth (Fig. 7). Understanding the relationship between $pCO_2$ and Miocene climate variability remains, however, a key challenge, as current $pCO_2$ reconstructions are still controversial, due to calibration issues, limited knowledge of past seawater chemistry, sparse data coverage and poor stratigraphic constraint[9,80,81]. A recent study nevertheless supports that $CO_2$ forcing due to increased tectonic degassing of carbon from higher ocean crustal production rates acted as a prime driver of MCO peak warmth[10]. $CO_2$ input from the intense phase of Columbia River magmatism between ~16.9 and 16 Ma may have additionally contributed to global warming during the initial phase of the MCO[17,19]. Modeling experiments further indicated that the elevated high-latitude temperatures and the low equator-to-pole temperature gradient during the MCO must have been driven by high $pCO_2$[16]. The parallel evolution of high-resolution upper ocean temperatures[34] and the marine $\delta^{18}O$ curve across the MCO and MMCT additionally supports that $pCO_2$ variations exerted a prime control on climate evolution. However, Miocene $pCO_2$ levels and variability during the MCO and MMCT remain contentious, and a deeper understanding of the proxies and of the climate feedbacks that may have amplified or dampened $pCO_2$ variations is urgently needed. Climate simulations that include higher variability of the ice sheet and atmospheric $pCO_2$ boundary conditions are also required to better understand their impact on Miocene heat budgets and the overturning circulation[16].

Our compilation of new and published data suggests that the production and export of southern-sourced deep water (proto-AABW) were highly sensitive to Antarctic ice sheet fluctuations and attendant sea surface temperature and salinity feedbacks close to Antarctica during the MCO. Threshold responses to high-latitude climate variability during the warmer part of the MCO (~16.9–16 Ma) led to radically different modes of deep water formation. Northward export of proto-AABW intensified during colder intervals at eccentricity/obliquity minima, whereas deep water production declined when the latitudinal temperature gradient decreased, and Antarctica became almost ice-free during periods of peak warmth between ~16.9 and 16 Ma. Following the Mi2 glacial expansion (~16 Ma), southern-sourced, carbonate undersaturated deep waters expanded northward into the equatorial and southern Pacific Ocean during the remainder of the MCO. Stepwise glacial expansion and cooling during the MMCT promoted upper ocean stratification and the onset of a near-modern Indo-Pacific circulation.

Taken together, these findings suggest that the projected increase in Southern Ocean stratification and slowdown in the global overturning circulation[3,5,59] may represent a transitory response to future global warming. Sustained high-latitude warming and retreat of the Antarctic ice sheet may, in the longer term lead to a decrease in density stratification that will alter the structure of the global circulation by shifting the main overturning from intermediate to deeper waters. Our study further underscores the key role of internal feedback in driving changes in intermediate and deep water formation and potentially triggering re-organizations of the Pacific overturning circulation.

## Methods

### Composite depth scale and stratigraphic splice

This study is based on a composite sediment splice from two holes drilled with the APC and HAPC systems at IODP Site U1490 (05°48.95′N, 142°39.27′E, 2341 m water depth), located on the northern edge of the Eauripik Rise in the Caroline Basin north of Papua New Guinea[29] (Fig. 1). The upper Lower to Middle Miocene succession recovered at Site U1490 consists of foraminifer-nannofossil ooze with minor amounts of clay and biogenic silica[29]. Detailed lithological descriptions are provided in ref. 29. The original shipboard composite

depth scale and stratigraphic splice were revised based on X-ray fluorescence (XRF) core scanning measurements combined with physical property data (Supplementary Note 2). The revised splice provides a continuous sedimentary record between 18.2 and 12.5 Ma, except for a disturbed interval in Section 1 of Core U1490B-27F, corresponding to a gap of ~100 kyr.

### Benthic foraminifer stable isotopes

The Site U1490 composite sediment splice was sampled at ~4 cm intervals between Samples U1490A-29F-2W, 86–88 cm and U1490B-25H-5W, 38–40 cm (286.78–243.77 revised meters composite depth, r-mcd) and at ~8 cm intervals between Samples U1490A-24H-5W, 148–150 cm and U1490B-24H-2W, 46–48 cm (243.69–230.82 r-mcd). In most samples, we selected three to ten well-preserved specimens of the epibenthic species *Cibicidoides wuellerstorfi* and/or *Cibicidoides mundulus* from the size fraction >250 µm for stable isotope analysis. Paired measurements of middle Miocene samples previously indicated no significant offsets in $\delta^{18}O$ and $\delta^{13}C$ between *C. wuellerstorfi* and *C. mundulus*[82]. In a few samples, where *C. wuellerstorfi* and *C. mundulus* were rare, only 1–2 specimens were measured. In 68 samples, where these species were absent, we analyzed *Oridorsalis umbonatus* (46 samples) or *Rectuvigerina* sp. (14 samples) or *Nuttallides* sp. (8 samples).

To normalize values to *C. wuellerstorfi/C. mundulus*, we subtracted 0.30‰ from *Rectuvigerina* sp. $\delta^{18}O$, based on 27 paired measurements of *C. mundulus* and *Rectuvigerina* sp., mean offset: 0.30‰, SD: 0.19‰ (Supplementary Note 3, Supplementary Data File 2). We did not include $\delta^{13}C$ values of *Rectuvigerina* sp., as the $\delta^{13}C$ offsets between these species exhibit high variability, reflecting the mobile infaunal habitat of this taxon. We subtracted 0.22‰ from *O. umbonatus* $\delta^{18}O$ and added 0.73‰ to *O. umbonatus* $\delta^{13}C$, based on 30 paired measurements of *C. mundulus* and *O. umbonatus*, mean offset: 0.22‰, SD: 0.20‰ for $\delta^{18}O$ and mean offset: -0.73‰, SD: 0.19‰ for $\delta^{13}C$ (Supplementary Note 3, Supplementary Data File 3). We did not apply any correction to *Nuttallides* sp. $\delta^{18}O$ and $\delta^{13}C$, as the offsets between *C. mundulus* and *Nuttallides* sp. were relatively small (based on 6 paired measurements, mean offset: 0.14‰, SD: 0.09‰ for $\delta^{18}O$, and mean offset: 0.08‰, SD: 0.07‰ for $\delta^{13}C$).

Foraminifer tests were broken into large fragments and cleaned in ethanol in an ultrasonic bath, then dried at 40 °C. Samples were analyzed with a Finnigan MAT 253 mass spectrometer coupled online to a Carbo-Kiel IV device for automated $CO_2$ preparation from carbonate samples at the Leibniz Laboratory for Radiometric Dating and Stable Isotope Research, Christian-Albrechts-University Kiel. Samples were reacted by individual acid addition (99% $H_3PO_4$ at 75 °C). The external standard error is better than ±0.08‰ for $\delta^{18}O$ and ±0.05‰ for $\delta^{13}C$, based on laboratory-internal and international carbonate standards. Results were calibrated using the international carbonate isotope standards National Bureau of Standard (NBS) 19: +1.95‰ VPDB ($^{13}C$), −2.20‰ VPDB ($^{18}O$), and International Atomic Energy Agency (IAEA) 603: +2.46‰ VPDB ($^{13}C$), −2.37‰ VPDB ($^{18}O$) as well as the internal carbonate standards Hela1: +0.91‰ VPDB ($^{13}C$), +2.48‰ VPDB ($^{18}O$), HB1: −12.10‰ VPDB ($^{13}C$), −18.10‰ VPDB ($^{18}O$) and SHK: +1.74‰ VPDB ($^{13}C$), −4.85‰ VPDB ($^{18}O$). Results are reported on the Vienna PeeDee Belemnite (VPDB) scale.

### Chronology

For Site U1490, orbital tuning was carried out by correlating the benthic foraminifer $\delta^{18}O$ and $\delta^{13}C$ records to a combination of eccentricity, obliquity, and precession parameters in the La04 astronomical solution[43] (Supplementary Note 2). As the primary tuning target, we used an eccentricity-tilt composite with no phase shift and an equal weight of eccentricity (E) and obliquity (T) in radians to which we added 0.2 of the precession parameter P (ET + 0.2P). Tie points are based on the correlation of prominent $\delta^{18}O$ minima to ET + 0.2P

maxima (Supplementary Table S2-2). The higher resolution of the Site U1490 isotope records than at the standard reference Site U1337[8,36,79] enabled refinement to the isotope chronology across the interval of low amplitude short eccentricity and high amplitude obliquity forcing between ~16.5 and 16.9 Ma (Supplementary Note 2).

Supplementary Note 2 provides detailed information concerning the chronology of Sites 1146, 1236, 1237, U1338, U1443, and Hole 751A. Stable isotope data on revised age models for Sites 1146, 1236, 1237, U1338, U1443, and Hole 751A are provided in Supplementary Data Files 6, 7, 10-13. We slightly modified the Site U1337 chronology in refs. [36,79]. by introducing an additional tie point at 371.61 m (r-mcd) that corresponds to the ETP maximum at 16,698 ka (Supplementary Information 2). We revised the chronology of Holes 761B and 1171B by correlating the published stable isotope records[14,46,83] to the U1338–U1337 composite chronology (Supplementary Note 2, Supplementary Data Files 8, 14).

### XRF scanning
We scanned core sections from along the shipboard splice at 9 and 50 kV on a 3rd generation Avaatech core scanner at the IODP Gulf Coats Repository at Texas A&M University. The XRF is equipped with a 100 W rhodium side-window X-ray tube. Prior to scanning, cores were moved from cold storage to the lab to warm to room temperature for at least 3 h to avoid condensation. Each section was cleaned from bottom to top using a glass microscope slide to remove the upper ~2 mm of sediment, and then covered with 4 mm-thick Ultralene film (SPEX Centriprep, Inc.) to protect the X-ray detector. Each section was scanned at 2 cm resolution along the center of the core section using a 1.2 cm$^2$ illumination window (1 cm downcore, 1.2 cm crosscore). Two successive scans were performed on each core section using different excitation settings. The first scan was performed at 9 kV, 0.25 mA, and no filter with 6 s of live time to analyze for Al, Si, K, Ca, Ti, and Fe. The second scan used 50 kV, 1.5 mA, and Cu filter with 10 s of live time to analyze for Sr and Ba. The raw X-ray spectra were processed using bAxil (Brightspec), which uses a nonlinear, least-squares method.

### Calibration of XRF scanner-derived carbonate content
We measured the calcium carbonate content of 33 discrete bulk sediment samples with a "Karbonat-Bombe", which has an instrumental error of ±1.0%[84]. Measurements were carried out on selected samples corresponding to XRF-scanner Ca maxima and minima over the entire investigated interval (Supplementary Note 4, Supplementary Data File 4). To calculate CaCO$_3$ weight% from the XRF data, a linear regression between the CaCO$_3$ measurements and the average of three consecutive XRF-scanner Log(Ca/(Al+Si)) values, which corresponds to the 2 cm range of each discrete sample (Supplementary Data File 5). The linear regression between the discrete measurements and Log(Ca/(Al + Si)) has an $R^2$ of 0.85 ($R$ = 0.92, $n$ = 33, Supplementary Note 4), which is in the range of calibrations at Sites U1337 and U1338 ($R^2$ = 0.87, $n$ = 255, ref. [85]) and Site U1443 ($R^2$ of 0.81, $n$ = 58, ref. [86]). Carbonate concentrations at Site U1490 range between 82% and 97%, corresponding to Log(Ca/(Al + Si)) values between 1.4 and 2.5 (Supplementary Data File 5). Carbonate concentrations below ~86% are associated with volcanic tephra layers and are not representative of the foraminifer-nannofossil ooze at Site U1490.

### Coarse (>63 μm) fraction percentage as a proxy for carbonate dissolution and carbonate ion concentration
Following the approach detailed in ref. [87], we used the weight% of the >63 μm coarse fraction residues as an indicator for carbonate dissolution at Site U1490. Residues >63 μm consist almost exclusively of biogenic carbonate (mainly planktic foraminifers with variable amounts of radiolarian tests in the 63–150 μm fraction). We calculated the weight percentage of the coarse fraction as a proportion of the total carbonate of the bulk sediment (estimated from calibrated XRF scanner Log(Ca/Si + Al) data) as follows:

Coarse fraction weight% of carbonate = (%CaCO$_3$ > 63 μm fraction/ %CaCO$_3$ total)×100

A brief discussion of previous applications and limitations of this method is provided in Supplementary Note 5.

### XRF scanner-derived Log(Si/Ti) as proxy for biogenic silica
We used the XRF scanner-derived Log(Si/Ti) data as a proxy for biogenic silica (opal) concentrations following the approach detailed in ref. [71]. Details are provided in Supplementary Note 6.

### Wavelet spectral analysis
Wavelet spectral analysis was performed on the evenly spaced time series of the residue weight% (Fig. 3), benthic $\delta^{18}O$ (Supplementary Figs. S2-4c and S3-6a), orbital eccentricity (Supplementary Fig. S3-6b) and XRF scanner-derived Log(Si/Ti) (Supplementary Fig. S6-1). Raw data were sampled at 1 kyr resolution by linear integration using the resampling function in AnalySeries 2.08[88]. The wavelet transform algorithm for evenly spaced time series analysis in 1 kyr temporal resolution was used with a Morlet basis function in the Past 4.10 software[89]. The $p$ = 0.05 significance level derived by a chi-square test is displayed as a black contour line and a cone of influence indicates the area with possible boundary effects[90].

## Data availability
The data sets generated and analyzed in this study are provided in Supplementary Data Files 1-14 and have been deposited at the Data Publisher for Earth & Environmental Science PANGAEA (https://doi.org/10.1594/PANGAEA.971967).

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

## Acknowledgements

This research used samples and data provided by the International Ocean Discovery Program. We thank the captain, crew, technical staff, and shipboard scientific party of IODP Expedition 363 for all their efforts and dedication. This project was funded by the Deutsche Forschungsgemeinschaft (DFG grant Ku649/40-1, 2 to W.K.).

## Author contributions

A.H. and W.K. conceived the study and performed data analyses. A.H., J.L., and N.A. generated the benthic foraminifer isotope data; D.K. generated the X-ray fluorescence (XRF) core scanning data and revised the original shipboard composite depth scale; A.H. and W.K. generated the carbonate data. D.K., G.M., Y.R., T.S., J.L., and

N.A. contributed ideas and critically discussed results. A.H. and W.K. drafted the manuscript and figures with contributions from D.K., G.M., Y.R., T.S., J.L., and N.A.

## Funding

## Competing interests

The authors declare no competing interests.
