## [Peer Review File · Nature Communications]

Re-organization of Pacific overturning circulation across the Miocene Climate OptimumREVIEWER COMMENTS

Reviewer #1 (Remarks to the Author):

The manuscript by Holbourn et al., entitled "Re-organization of Pacific overturning circulation across the Miocene Climate Optimum", focuses on the change of deep-water sources in the western equatorial Pacific Ocean. It provides high-resolution stable isotope data spanning the interval 18.2 – 12.5 Ma from the west equatorial Pacific Ocean. The author compares their stable isotope records with other key locations in the Pacific, Indian, and Southern Oceans, revealing the shift in southern sourced water. This manuscript also provides valuable astronomically tuned stable isotopic data that covered two important events during the Early Miocene.

Overall, the manuscript could benefit from some clarification in places to strengthen its message (see below). Irrespective, this work suits for Nature Communications.

1. This manuscript discusses the re-organization of Pacific overturning circulation during the Miocene Climate Optimum (MCO). The author briefly introduces the Early Miocene climate and major climatic events. However, the main text lacks an introduction to the Pacific overturning, making it difficult to fully understand the study. I noticed that the author discusses the modern Pacific Overturning in the supplemental materials. Perhaps, it would be beneficial to mention it in the Introduction as well?
2. Although the title of this paper is "Re-organization of Pacific overturning circulation across the Miocene Climate Optimum", the discussion mainly focuses on the southern sourced deep water. I would expect more information and discussion about the evolution and response to the climate change of the entire overturning system throughout the Early Miocene after reading the title of this manuscript but the content was mainly focused on the limb of the Pacific overturning circulation.

Below are some minor suggestions.

Introduction

Line 76, could you add a reference for MMCT?

Results and Discussion

Line 110, please add references for the 'Previous studies'

Line 115, change 'record' to 'records'

Lines 141-142, please refer to a figure here for the description.

Lines 165-166, could you explain the 'new circulation patterns' after mentioning it in the text?

Lines 170-173, are there any references or figures that can be cited for this argument?

Line 183, could you provide a few examples of 'robust species' here?

Lines 186-189, the sediment coarse fraction $>63\mu\text{m}$ weight percentage can also reflect the current energy/current winnowing, however, there was no discussion about the current energy in the text.

Lines 216-221, the $\delta^{13}\text{C}$ gradients vary between 16.9 – 16 Ma and do not show a very clear decreasing pattern, it is a little bit farfetched to describe it as a decreasing trend. Could you calculate the gradients between each site and show them in the supplemental material to prove that they are all exhibiting a similar trend? Moreover, this sentence is quite long and contains many messages, will it be better to separate it into a few sentences to keep the message clear? The reference figure(s) for the changes in the $\delta^{13}\text{C}$ gradients is(are) missing.

Line 221, could you explain what is the shallow $\delta^{13}\text{C}$ gradient? Maybe you mean the $\delta^{13}\text{C}$ gradient from shallow sites?

Line 245, Fig.6 is a nice summary of the Pacific overturning circulation, could you discuss more about this figure?

Lines 322 – 323, what do diverse and well-preserved planktic foraminifera assemblages indicate?

Line 324, could you give a few examples of 'smaller, fragile, dissolution-prone species'?

Lines 329-332, are there any studies that can support your speculation?

Lines 342-344, what pattern(s) of $\delta^{13}\text{C}$ indicate the stepwise cooling and expansion of the Antarctic ice sheet during the MMCT?

Figures

Figure 5. The tick marks on the left-hand side of panels A and B were shifted, please move them to the correct places.

Dataset

The unit (kyr) of age at the last column in the file named 476349_0_data_set_8479709_s69pz5 is missing.

Supplement

The correlation between the two holes (A and B) was done by comparing XRF scanning and physical properties data. In Fig S2-1, the revised composite scale looks convincing but the revised data between 275.6 – 279 r-mcd is less satisfying. The original data (Fig. S2-2A) correlate with each other much better than the revised data. Therefore, adding another tie point to this interval may provide a better solution.

Generally, the age-depth tuning approach is thoroughly described. The author explained their tuning procedure in the supplemental materials that they tuned the $\delta^{18}\text{O}$ minima to maxima of eccentricity-obliquity composite between 18.2 – 12.4 Ma. However, the original stable isotope records were in the depth domain. Were the age of 12.4 Ma and 18.2 coming from the tie points? Could you mention it in the text? Meanwhile, there was no information on why you tuned the $\delta^{18}\text{O}$ minima to the maxima of the eccentricity-obliquity composite, could you explain why you chose to tune the $\delta^{18}\text{O}$ minima to the astronomical parameter maxima or add some references for it?

Reviewer #2 (Remarks to the Author):

Review of manuscript NCOMMS-476349 "Re-organization of Pacific overturning circulation across the Miocene Climate Optimum" by Ann Holbourn, Wolfgang Kuhnt, Denise K. Kulhanek, Greg Mountain, Yair Rosenthal, Takuya Sagawa, Julia Lübbers, Nils Andersen

The authors present a new $\delta^{13}\text{C}$ and $\delta^{18}\text{O}$ record, supplemented by XRF-derived elemental data and sedimentary characteristics from the IODP Site U1490 in the western equatorial Pacific to investigate the evolution of deep water circulation during the Miocene Climate Optimum (MCO) and Middle Miocene Climate Transition (MMCT). By comparing their record to other available records from the Pacific, Indian and Southern Oceans the authors track the changes in the water mass characteristics during the MCO and MMCT.

This study presents an exciting, high-quality record that significantly improves our understanding of the ocean and climatic changes during one of the key intervals of Earth's history. The authors should be lauded for their work and efforts, and I have no doubt that this record and study will be of great interest and influence to the paleoceanographic community.

Personally, given the largely descriptive nature of this manuscript and the many interesting findings and features of their record that are difficult to extract in a concise manner and would benefit from more detailed explanation I would not have opted for a short style journal format such as Nat. Comm. for this work, but I leave that decision to the authors and the Editors.

This manuscript is overall well-written and well-illustrated. My report is rather short; I do however

have a couple of main points that I would like to ask the authors to address before this can be considered for publication, which I summarize in the following. I provide more specific comments on the manuscript and supplementary materials further below.

Firstly, the authors interpret their data in context of available CO₂ derived from boron isotopes. I have no doubts that the boron isotope data are of good quality, but given the scarcity of the data and the uncertainty in $\delta^{11}\text{B}_{\text{sw}}$ composition at this time that is critical for robust CO₂ calculation from Miocene foraminifers' $\delta^{11}\text{B}$ values I would be more careful when making comparisons between the authors' data and the available CO₂. Also, there is a large variability in CO₂ during the early part of the MCO (orange) which disappears from the CO₂ record afterwards. The authors state in lines 359-360 "Our results highlight the sensitivity of the Pacific overturning circulation to variations in atmospheric CO₂ and ice volume on a warmer Miocene Earth (Figure 7)". I might be missing something, but I struggle to see this connection. Could you elaborate on the variability and features found in your record and how they compare to the variable CO₂ during the initial (orange) phase of the MCO but not afterwards? What site does the CO₂ data come from and how does it compare to U1490? Also, the peaks in the CO₂ data looks to me offset from the peaks in the $\delta^{13}\text{C}$ gradient. This might be real, but have you checked the CO₂ data age model? Is it same as your age model?

The authors briefly mention the potential role of the CRB as the prime driver of the climate evolution during the MCO and MMCT, and/or potentially crustal production rates. Based on their findings, what can the authors say about the main climate driver (and maybe more specifically CO₂ and temperatures that have been so hard to reconcile thus far for this time period)?

Secondly, this manuscript is in essence descriptive with conclusions sustained by inferences. To me, at least in the form the findings are currently presented the manuscript calls for modelling and more quantitative constraints. I am sorry to be asking this as I am aware this will present some extra work to the authors, however; in my opinion this would substantially improve the manuscript and strengthen the authors' conclusions. It would be interesting to see the evolution of the deep water chemistry across the MCO and the posterior emergence of the modern-like Indo-Pacific circulation in response to the MMCT cooling explored in a modelling framework with representative thermohaline circulation. For example, can such modelling more convincingly explain and track the contraction/expansion of bottom water saturation in response to Antarctic ice sheet fluctuations? Can it provide a compelling mode for Pacific overturning during the MCO based on the proxy records? What is the underlying driving mechanism of the circulation reorganisation? The inclusion of modelling might also further aid the authors better extract their main findings and make their study more accessible in a short paper format.

Specific comments:

Line 307: formatting

Lines 329-332: In lines 289-291, the authors suggest that the expansion of the Mi2 ice sheet promoted the expansion of corrosive deep waters. Yet here the authors suggest that the global cooling during the MMCT would lead to increase in carbonate ion concentration of deep waters. Why the different effects of cooling? How do the authors explain this?

Lines 401-419: It would be great to see how this in a figure – could you add a figure to the Supplement which compares the $\delta^{13}\text{C}$ and $\delta^{18}\text{O}$ of the different foram species and illustrates the corrections applied?

Line 425: Please provide the mean value of the internal standard and external carbonate reference

materials measured.

Line 501-503: Of what? Please specify.

Figure 1: I would suggest distinguishing the site U1490 from the other sites, by using different marker colour fill for example.

Figure 2: I am curious about the sharp decrease in CaCO₃ weight % at ~15.2 Ma. What is it, is this a known event?

Supplementary Information

Figure S1-1. Would be helpful to label the different water masses/currents in the figure.

Figure S1-2 and S1-2: These figures are very useful – does the distribution of $\delta^{13}C$, alkalinity and dissolved silicon look similar? Could you include them in figures as well?

Supplementary Information 2: Site U1490 revised composite depth scale and stratigraphic splice – I really appreciated this section. Would add to Fig. S2-1 mention of why Ba/Sr was used.

Table S2-2: Would suggest to also spell out again what r-mcd is in the caption.

Supplementary Data files – are overall very good and helpful. Please add foram species to "476349_0_data_set_8479709_s69pz5". Could you also provide the literature data from Figures S7-2 and S7-3 and Figure 5 in the main manuscript so that these figures can be easily redrawn by readers?

Reviewer #3 (Remarks to the Author):

The manuscript by Holbourn and co-authors presents a nice and well-written study. This is a great dataset and a sound interpretation of the data and for sure something that should be published in a journal like Nature Communications.

Still I have a couple of comments and suggestions that I hope make the manuscript better and in parts also better understandable for the reader.

These are in the order they appear in the manuscript:

- Line 86: what is the estimated paleo water depth?
- Lines 144-148 and also line 175: Here I do not see why the authors argue for a larger ice sheet after Mi-2. For me the dataset in figure 2 suggests that there is a warming trend before MI-2 (at the onset of MCO) and the Mi-2 glaciation is just bringing the system back to 'normal' i.e. the values (and therefore in the interpretation of the $d_{18}O$ as an ice proxy) before the MCO warming.
- Lines 170-174: This dissolution interval raises the question about preservation changes across the study interval. How good/bad are the foraminiferal shells preserved across the entire record and also across glacial/interglacial changes? This is something that should be discussed and shown to allow a judgement about the quality of the data.
- Line 229: There is no $d_{18}O$ shown in figure 5
- Chapter starting at line 212: This chapter is internally consistent in its interpretation of the data. But I miss a clear and convincing statement at the beginning that all of these data reflect SCW and not another water mass like North pacific waters which might still be different from PIW at Site 1146 (see e.g. studies of Burls et al). As a reader this chapter seems to build on the assumption that Site U1490 is bathed by SCW but without explanation or justification.

- Line 274-275: sorry, but I do not see the proposed relation between carbonate-depleted and colder periods in the interval between 16.9 and 16.2 Ma in figure 3.
- Line 290ff: but CaCO₃ is decreasing well before Mi-2! So how reliable is the stated speculation?
- Line 319: where is the entire Log Si/Ti record shown? Would be good to have this is one of the main figures.
- Figure 5: why is Site 1146 in panel d not plotted against the Site U1490 record as all the other sites?

ORIGINAL MANUSCRIPT NCOMMS-23-63602A

REVIEWER COMMENTS

Reviewer #1 (Remarks to the Author):

The manuscript by Holbourn et al., entitled "Re-organization of Pacific overturning circulation across the Miocene Climate Optimum", focuses on the change of deep-water sources in the western equatorial Pacific Ocean. It provides high-resolution stable isotope data spanning the interval 18.2 – 12.5 Ma from the west equatorial Pacific Ocean. The author compares their stable isotope records with other key locations in the Pacific, Indian, and Southern Oceans, revealing the shift in southern sourced water. This manuscript also provides valuable astronomically tuned stable isotopic data that covered two important events during the Early Miocene. Overall, the manuscript could benefit from some clarification in places to strengthen its message (see below). Irrespective, this work suits for Nature Communications.

We thank Reviewer 1 for detailed, critical comments that considerably helped us to improve the manuscript.

1. This manuscript discusses the re-organization of Pacific overturning circulation during the Miocene Climate Optimum (MCO). The author briefly introduces the Early Miocene climate and major climatic events. However, the main text lacks an introduction to the Pacific overturning, making it difficult to fully understand the study. I noticed that the author discusses the modern Pacific Overturning in the supplemental materials. Perhaps, it would be beneficial to mention it in the Introduction as well?

We have revised the Introduction following the reviewer's advice (Lines 82-97). We included an outline of present-day Pacific overturning circulation, referring to Supplementary Information 1, which presents a more detailed overview of the modern overturning circulation including additional figures.

2. Although the title of this paper is "Re-organization of Pacific overturning circulation across the Miocene Climate Optimum", the discussion mainly focuses on the southern sourced deep water. I would expect more information and discussion about the evolution and response to the climate change of the entire overturning system throughout the Early Miocene after reading the title of this manuscript but the content was mainly focused on the lower limb of the Pacific overturning circulation.

Our study focuses on the lower limb of the Pacific overturning circulation, since Site U1490 is located in a unique position to monitor the deep water inflow from the Southern Ocean along the western Pacific margin. Today, these southern-sourced waters play a crucial role as a reservoir of heat, nutrients and carbon, thus, exerting a major control over the entire Pacific overturning circulation. Our results suggest that the southern-sourced inflow was also the main driver of Pacific overturning on a unipolar glaciated Earth during the MCO and MMCT, although ocean chemistry and intermediate/deep water production fundamentally changed in response to Antarctic ice sheet fluctuations and variations in the latitudinal thermal gradient. Given the crucial role of southern-sourced waters for the entire Pacific overturning circulation, we felt that the title of the paper was suitable.

We discussed the evolution and response of the Pacific overturning upper limb to climate change during the MCO and MMCT in two sections of the manuscript: (1) we suggested that the overturning upper limb became less important during the MCO due to weaker westerlies and reduced wind-driven upwelling, as indicated in a recent modelling study by Frigola et al., 2021 (Lines 307-315; Figure 6); (2) we argued that a more vigorous overturning upper flow became established during the MMCT, signaling the onset of a near modern overturning circulation (Lines 400-403, Figure 6).

References

Frigola, A., Prange, M., & Schulz, M. A dynamic ocean driven by changes in CO₂ and Antarctic ice-sheet in the middle Miocene. *Palaeogeography, Palaeoclimatology, Palaeoecology*, 579, 110591 (2021).
<https://doi.org/10.1016/j.palaeo.2021.110591>

Below are some minor suggestions.

Introduction

Line 76, could you add a reference for MMCT?
Done (Line 75).

Results and Discussion

Line 110, please add references for the 'Previous studies'
Done (Line 135).

Line 115, change 'record' to 'records'
Done (Line 138).

Lines 141-142, please refer to a figure here for the description.
Done (Line 165).

Lines 165-166, could you explain the 'new circulation patterns' after mentioning it in the text?
Done (Lines 190-193).

Lines 170-173, are there any references or figures that can be cited for this argument?
We now refer to Figures 2-3, panels b-d.

Line 183, could you provide a few examples of 'robust species' here?
Done (Lines 209-212).

Lines 186-189, the sediment coarse fraction >63 μ m weight percentage can also reflect the current energy/current winnowing, however, there was no discussion about the current energy in the text.

To clarify this point, we added the following text to Supplementary Information 5:
"We did not relate the variability in the >63 μ m residue percentages to changes in bottom water current intensity, since the fine fraction <63 μ m removed by winnowing would have resulted in reduced carbonate accumulation rates, which is not evident in our records (Supplementary Figure S5-1). For example, carbonate accumulation rates were generally elevated between 13.8 and 13.6 Ma and between 17.4 and 17.8 Ma, when >63 μ m residue percentages were high. "

Lines 216-221, the $\delta^{13}\text{C}$ gradients vary between 16.9 – 16 Ma and do not show a very clear decreasing pattern, it is a little bit farfetched to describe it as a decreasing trend. Could you calculate the gradients between each site and show them in the supplemental material to prove that they are all exhibiting a similar trend? Moreover, this sentence is quite long and contains many messages, will it be better to separate it into a few sentences to keep the message clear? The reference figure(s) for the changes in the $\delta^{13}\text{C}$ gradients is(are) missing.

Thank you for pointing this out. We reviewed our interpretation and revised the text and Figure 5: (1) we no longer refer to a progressive decrease in the $\delta^{13}\text{C}$ gradients between 16.9 and 16 Ma (Lines 255-260); (2) we added an additional panel to Figure 5, which more clearly delineates the evolution of $\delta^{13}\text{C}$ gradients (see Figure 5B reproduced below).

Figure 5B (next page): **B.** Evolution of benthic foraminifer $\delta^{13}\text{C}$ gradients from **a** Sites U1490 and 1237, **b** Sites U1490 and U1443, **c** Sites U1490 and U1337-U1338, **d** Sites U1490 and 1146, **e** Sites U1337-U1338 and 1146, **f** Site U1490 benthic foraminifer $\delta^{18}\text{O}$ (3 point running mean). Monterey Excursion carbon isotope maxima^{82,94} are labeled CM1, 2, 3, 4a, 4b, 5a, 5b, and 6. Warmer phase of MCO from ~16.9 to 16 Ma shaded dark orange; hyperthermal phase of MCO following Mi2 glaciation shaded light orange. Expansion of Antarctic ice sheet following Mi3 shaded light blue.

Line 221, could you explain what is the shallow $\delta^{13}\text{C}$ gradient? Maybe you mean the $\delta^{13}\text{C}$ gradient from shallow sites?

We replace “shallow” by “weak” to clarify this point (Line 249).

Line 245, Fig.6 is a nice summary of the Pacific overturning circulation, could you discuss more about this figure?

We re-organized the section and expanded the text to better contrast the modern and MCO scenarios presented in Figure 6 (Lines 292-303).

Lines 322 – 323, what do diverse and well-preserved planktic foraminifera assemblages indicate?

We revised the sentence to explain the significance of these better-preserved assemblages. (Lines 363-368).

Line 324, could you give a few examples of ‘smaller, fragile, dissolution-prone species’?

We provided a few examples on Lines 363-368.

Lines 329-332, are there any studies that can support your speculation?

We now refer to observations and model-data assimilations that show that the modern Southern Ocean is highly sensitive to changes in atmospheric $p\text{CO}_2$ and plays a key role in regulating oceanic CO_2 uptake and carbon storage (DeVries, 2014; Long et al., 2021; Gruber, et al., 2023). See Lines 376-378.

References:

DeVries, T. (2014), The oceanic anthropogenic CO_2 sink: Storage, air-sea fluxes, and transports over the industrial era, *Global Biogeochem. Cycles*, 28, 631–647, doi:10.1002/2013GB004739

Long, M.C. et al., 2021. Strong Southern Ocean carbon uptake evident in airborne observations, *Science* Vol 374, Issue 6572 pp. 1275-1280, doi: 10.1126/science.abi435

Gruber, N. et al. Trends and variability in the ocean carbon sink. *Nat. Rev. Earth Environ.* 4, 119–134 (2023). <https://doi.org/10.1038/s43017-022-00381-x>

Lines 342-344, what pattern(s) of $\delta^{13}\text{C}$ indicate the stepwise cooling and expansion of the Antarctic ice sheet during the MMCT?

We modified the text to clarify that our interpretation is based on comparisons of both $\delta^{18}\text{O}$ and $\delta^{13}\text{C}$ profiles (Lines 388-397).

Figures

Figure 5. The tick marks on the left-hand side of panels A and B were shifted, please move them to the correct places

Done.

Dataset

The unit (kyr) of age at the last column in the file named 476349_0_data_set_8479709_s69pz5 is missing.

Done.

Supplement

The correlation between the two holes (A and B) was done by comparing XRF scanning and physical properties data. In Fig S2-1, the revised composite scale looks convincing but the revised data between 275.6 – 279 r-mcd is less satisfying. The original data (Fig. S2-2A) correlate with each other much better than the revised data. Therefore, adding another tie point to this interval may provide a better solution.

Thank you for pointing out this inconsistency, which stemmed from a drafting error. We have now amended the supplementary figure and provided the correct tie points between Holes U1490A and B. The revised figure shows that there is a good match between the records from the two holes.

Generally, the age-depth tuning approach is thoroughly described. The author explained their tuning procedure in the supplemental materials that they tuned the $\delta^{18}\text{O}$ minima to maxima of eccentricity-obliquity composite

between 18.2 – 12.4 Ma. However, the original stable isotope records were in the depth domain. Were the age of 12.4 Ma and 18.2 coming from the tie points? Could you mention it in the text? Meanwhile, there was no information on why you tuned the $\delta^{18}\text{O}$ minima to the maxima of the eccentricity-obliquity composite, could you explain why you chose to tune the $\delta^{18}\text{O}$ minima to the astronomical parameter maxima or add some references for it?

The approximate ages for the tuned interval were initially based on the shipboard biostratigraphy (Rosenthal et al., 2018). These ages were later revised following orbital tuning. We added a sentence in Supplementary Information 2 to clarify this point.

We followed the suggestion from the reviewer and explained the reasoning behind the tuning of $\delta^{18}\text{O}$ minima to maxima of the eccentricity-obliquity composite in the Supplementary Information 2. We followed the strategy previously outlined in Holbourn et al. (2005): “We tuned $\delta^{18}\text{O}$ minima to obliquity maxima, since we assumed (1) that relatively warm summers during high obliquity would promote ice-sheet melting in Antarctica, whereas cool summers during low obliquity would favour ice-sheet growth, and (2) that a low summer insolation gradient between low and high latitudes during high obliquity would decrease poleward moisture transport, inhibiting ice-sheet build-up”. Increased high latitude precessional summer insolation during eccentricity maxima would additionally favour the formation of warmer deep water at high latitudes and promote Antarctic ice melt, thus, increasing transfer of ^{16}O from the Antarctic ice sheet to the ocean.

Reviewer #2 (Remarks to the Author):

Review of manuscript NCOMMS-476349 “Re-organization of Pacific overturning circulation across the Miocene Climate Optimum” by Ann Holbourn, Wolfgang Kuhnt, Denise K. Kulhanek, Greg Mountain, Yair Rosenthal, Takuya Sagawa, Julia Lübbbers, Nils Andersen

The authors present a new $\delta^{13}\text{C}$ and $\delta^{18}\text{O}$ record, supplemented by XRF-derived elemental data and sedimentary characteristics from the IODP Site U1490 in the western equatorial Pacific to investigate the evolution of deep water circulation during the Miocene Climate Optimum (MCO) and Middle Miocene Climate Transition (MMCT). By comparing their record to other available records from the Pacific, Indian and Southern Oceans the authors track the changes in the water mass characteristics during the MCO and MMCT.

This study presents an exciting, high-quality record that significantly improves our understanding of the ocean and climatic changes during one of the key intervals of Earth's history. The authors should be lauded for their work and efforts, and I have no doubt that this record and study will be of great interest and influence to the paleoceanographic community.

Personally, given the largely descriptive nature of this manuscript and the many interesting findings and features of their record that are difficult to extract in a concise manner and would benefit from more detailed explanation I would not have opted for a short style journal format such as Nat. Comm. for this work, but I leave that decision to the authors and the Editors.

This manuscript is overall well-written and well-illustrated. My report is rather short; I do however have a couple of main points that I would like to ask the authors to address before this can be considered for publication, which I summarize in the following. I provide more specific comments on the manuscript and supplementary materials further below.

We thank Reviewer 2 for detailed, critical comments that considerably helped us to improve the manuscript.

Firstly, the authors interpret their data in context of available CO_2 derived from boron isotopes. I have no doubts that the boron isotope data are of good quality, but given the scarcity of the data and the uncertainty in $\delta^{11}\text{B}_{\text{sw}}$ composition at this time that is critical for robust CO_2 calculation from Miocene foraminifers' $\delta^{11}\text{B}$ values I would be more careful when making comparisons between the authors' data and the available CO_2 . Also, there is a large variability in CO_2 during the early part of the MCO (orange) which disappears from the CO_2 record afterwards. The authors state in lines 359-360 “Our results highlight the sensitivity of the Pacific overturning circulation to variations in atmospheric CO_2 and ice volume on a warmer Miocene Earth (Figure 7)”. I might be missing something, but I struggle to see this connection. Could you elaborate on the variability and features found in your record and how they compare to the variable CO_2 during the initial (orange) phase of the MCO but not afterwards?

We agree that a direct link between pCO_2 and ice volume variations cannot be demonstrated, based on current pCO_2 reconstructions over the MCO and MMCT interval. Miocene pCO_2 reconstructions remain controversial due to various issues including proxy calibration, limited knowledge of past seawater chemistry and data coverage. Data coverage is especially sparse over the final part of the MCO and over the MMCT (Figure 7). Furthermore, pCO_2 reconstructions are generally from sites, which have relatively low sedimentation rates or discontinuous records (hiatuses, poor core recovery, etc..) and lack a well-constrained isotope cyclo-stratigraphy. Evidence for increased ocean crustal production rates (Herbert et al., 2022) and increased volcanism during the initial warmer part of the MCO (Kasbohm et al., 2023) as well as modelling experiments (Burls et al., 2021; Frigola et al., 2021) nevertheless support that CO_2 forcing was a prime driver of climate evolution across the MCO. To address the reviewer's criticism, we have revised the abstract (Line 35) and expanded the discussion (Lines 406-425), acknowledging the limitations of Miocene pCO_2 reconstructions and highlighting the needs for a deeper understanding of the proxies and for relevant modelling studies to better understand climate dynamics on a warmer Earth.

References:

- Burls, N. J., Bradshaw, C. D., De Boer, A. M., Herold, N., Huber, M. et al. (2021). Simulating Miocene warmth: Insights from an opportunistic multi-model ensemble (MioMIP1). *Paleoceanography and Paleoclimatology*, 36, e2020PA004054. <https://doi.org/10.1029/2020PA004054>
- Frigola, A., Prange, M., and Schulz, M. (2021), A dynamic ocean driven by changes in CO_2 and Antarctic ice-sheet in the middle Miocene. *Palaeogeography, Palaeoclimatology, Palaeoecology*, 579, 110591
- Herbert, T.D. et al. Tectonic degassing drove global temperature trends since 20 Ma. *Science* 377, 116–119 (2022). <https://doi.org/10.1126/science.abl4353>
- Kasbohm, J. et al. Eruption history of the Columbia River Basalt Group constrained by high-precision U-Pb and $^{40}\text{Ar}/^{39}\text{Ar}$ geochronology. *Earth Planet. Sci. Lett.* 617, 118269 (2023). <https://doi.org/10.1016/j.epsl.2023.118269>

What site does the CO_2 data come from and how does it compare to U1490?

Also, the peaks in the CO₂ data looks to me offset from the peaks in the $\delta^{13}\text{C}$ gradient. This might be real, but have you checked the CO₂ data age model? Is it same as your age model?

Thank you for pointing this out. The data shown on Figure 7 were from the pCO₂ compilation of Rae et al. 2021, based on the original published age models. Most of the boron isotope derived data shown on Figure 7 are from Hole 761B (Greenop et al., 2014; Foster et al., 2012). We have now revised Figure 7 and plotted the Site 761 pCO₂ data according to our revised age model for this site, which is fully compatible with the U1490 chronology (Supplementary Figure S7-3).

The 761 pCO₂ data on our revised age model indicate high variability during the initial warmer part of the MCO with estimates reaching 1200 ppm or more (Figure 7). However, the high pCO₂ variability during the early part of the MCO cannot be directly tied to high amplitude $\delta^{18}\text{O}$ climate cycles, due to the low sedimentation rates (mainly <0.5 cm/kyr) and consequent low resolution of the $\delta^{18}\text{O}$ record at Site 761 (Supplementary Figure S7-2). After 16 Ma, the boron isotope pCO₂ data are extremely sparse (Figure 7) and do not provide insight into variability during the final part of the MCO and MMCT. Summing up, understanding the relationship between pCO₂ and Middle Miocene climate variability remains a key challenge and new data from sites with well-constrained chronologies are urgently needed. We have discussed these issues in the revised manuscript (Lines 406-425, 973-976).

References:

- Foster, G.L., Lear, C.H., & Rae, W.B., The evolution of pCO₂, ice volume and climate during the middle Miocene. *Earth and Planetary Science Letters* 341-344, 243–254 (2012). <http://dx.doi.org/10.1016/j.epsl.2012.06.007>
- Greenop, R., Foster, G.L., Wilson, P.A. & Lear, C.H. Middle Miocene climate instability associated with high-amplitude CO₂ variability, *Paleoceanography*, 29, 845–853 (2014). doi:10.1002/2014PA002653
- Rae, J.W.B. *et al.* Atmospheric CO₂ over the Past 66 Million Years from Marine Archives. *Annu. Rev. Earth Planet. Sci.* 49, 609-641 (2021). <https://doi.org/10.1146/annurev-earth-082420-063026>

The authors briefly mention the potential role of the CRB as the prime driver of the climate evolution during the MCO and MMCT, and/or potentially crustal production rates. Based on their findings, what can the authors say about the main climate driver (and maybe more specifically CO₂ and temperatures that have been so hard to reconcile thus far for this time period)?

We followed the reviewer's suggestion and expanded the section entitled "Perspective" to discuss the primary drivers of middle Miocene climate (Lines 406-429).

Secondly, this manuscript is in essence descriptive with conclusions sustained by inferences. To me, at least in the form the findings are currently presented the manuscript calls for modelling and more quantitative constraints. I am sorry to be asking this as I am aware this will present some extra work to the authors, however; in my opinion this would substantially improve the manuscript and strengthen the authors' conclusions. It would be interesting to see the evolution of the deep water chemistry across the MCO and the posterior emergence of the modern-like Indo-Pacific circulation in response to the MMCT cooling explored in a modelling framework with representative thermohaline circulation. For example, can such modelling more convincingly explain and track the contraction/expansion of bottom water saturation in response to Antarctic ice sheet fluctuations? Can it provide a compelling mode for Pacific overturning during the MCO based on the proxy records? What is the underlying driving mechanism of the circulation reorganisation? The inclusion of modelling might also further aid the authors better extract their main findings and make their study more accessible in a short paper format.

We agree with the reviewer that a modelling study to explore changes in bottom water alkalinity during the MCO and the emergence of the modern-like Indo-Pacific circulation during the MMCT in response to Antarctic ice sheet fluctuations would be enormously beneficial to back up our interpretations. However, after seeking advice from specialists, we concluded that including a modelling component would be beyond the scope of the present manuscript. Given the current limited understanding of boundary conditions during the MCO and MMCT, a reliable modelling framework would require (and justify) in-depth evaluation of various pCO₂ and ice sheet scenarios to assess impacts on the overturning circulation.

In particular, simulating the low MCO meridional temperature gradient remains an outstanding challenge for most models (Burls et al., 2021). The low meridional temperature gradients suggested by proxy records are not well reproduced in recent coupled climate models that were run under ~400 ppm atmospheric pCO₂ boundary conditions (Goldner et al., 2014; Burls et al., 2021). Simulations with MCO vegetation and paleogeographic boundary conditions at 400 ppm pCO₂ (Frigola et al., 2021) do not reproduce observed ice free or reduced sea-ice conditions in the Ross Sea (Levy et al., 2016). More realistic gradients are achieved using higher pCO₂ conditions, but then the tropics become unusually warm. Climate simulations that include higher variability of ice sheet and pCO₂ boundary conditions are needed to fully understand the relative effects of the Antarctic ice sheet and atmospheric pCO₂ on the variability of Miocene heat budgets and overturning circulation (Burls et al., 2021). Adding a modelling component to our paper would require dedicated efforts and a considerable time investment in order to provide robust results. We therefore feel that it cannot be feasibly incorporated into the current paper, but definitely warrants an independent specialist investigation. In our revision, we nevertheless attempted to constrain our findings in the light of some recent modelling studies (Lines 270-274, 310-312, 415-417).

References:

- Burris, N. J., Bradshaw, C. D., De Boer, A. M., Herold, N., Huber, M. et al. (M. Pound, Y. Donnadieu, A. Farnsworth, A. Frigola, E. Gasson, A. S. von der Heydt, D. K. Hutchinson, G. Knorr, K. T. Lawrence, C. H. Lear, X. Li, G. Lohmann, D. J. Lunt, A. Marzocchi, M. Prange, C. A. Riihimaki, A.-C. Sarr, N. Siler, Z. Zhang) (2021). Simulating Miocene warmth: Insights from an opportunistic multi-model ensemble (MioMIP1). *Paleoceanography and Paleoclimatology*, 36, e2020PA004054. <https://doi.org/10.1029/2020PA004054>
- Frigola, A., Prange, M., and Schulz, M. (2021), A dynamic ocean driven by changes in CO₂ and Antarctic ice-sheet in the middle Miocene. *Palaeogeography, Palaeoclimatology, Palaeoecology*, 579, 110591, <https://doi.org/10.1016/j.palaeo.2021.110591>
- Goldner, A., Herold, N., and Huber, M. (2014), The challenge of simulating the warmth of the mid-Miocene climatic optimum in CESM1. *Clim. Past*, 10, 523–536, doi:10.5194/cp-10-523-2014
- Levy, R., Harwood, D., Florindo, F., Sangiorgi, F., Tripathi, R., von Eynatten, H., Gasson, E., Kuhn, G., Tripathi, A., DeConto, R., Fielding, C., Field, B., Golledge, N., McKay, R., Naish, T., Olney, M., Pollard, D., Schouten, S., Talarico, F., Warny, S., Willmott, V., Acton, G., Panter, K., Paulsen, T., Taviani, M., and SMS Science Team (2016), Antarctic ice sheet sensitivity to atmospheric CO₂ variations in the early to mid-Miocene, *P. Natl. Acad. Sci. USA*, 113, 3453–3458, <https://doi.org/10.1073/pnas.1516030113>, 2016.

Specific comments:

Line 307: formatting

data⁶¹ and X-ray fluorescence scanner-derived estimates of biogenic silica and carbonate
Done (Lines 348-349).

Lines 329-332: In lines 289-291, the authors suggest that the expansion of the Mi2 ice sheet promoted the expansion of corrosive deep waters. Yet here the authors suggest that the global cooling during the MMCT would lead to increase in carbonate ion concentration of deep waters. Why the different effects of cooling? How do the authors explain this?

Boundary conditions (e.g., ice volume, sea ice extent, atmospheric CO₂, latitudinal temperature gradients, strength of westerlies and Southern Ocean upwelling) differed markedly during the MCO and MMCT. During Mi2, the mean $\delta^{18}\text{O}$ seawater increase was $\sim 0.2\text{‰}$ seawater and ice expansion likely remained confined inland on Antarctica. By contrast, the EAIS increased by $\sim 0.5\text{‰}$ (equivalent to $\sim 50\text{ m}$ sea level fall) during Mi3, when pCO₂ and global temperatures decreased. During Mi3, the EAIS likely reached the coastline, favoring local sea ice development. These fundamental differences during Mi2 and Mi3 would have altered Southern Ocean stratification, intermediate/deep water formation processes and water mass properties in very different ways. The steeper latitudinal temperature gradient after Mi3 would have affected the strength of westerlies and the intensity of Southern Ocean upwelling, which plays a key role in regulating intermediate/deep water formation and the structure of the overturning circulation. Our results suggest that Mi2 led to the weakening of the Southern Ocean upper cell and expansion of corrosive deep waters (Lines 307-315). By contrast, Mi3 marked a major shift towards a modern Pacific circulation with intensification of the overturning circulation's upper limb and development of a more differentiated central Pacific water mass (close to the modern PCW) after $\sim 13.8\text{ Ma}$ (Lines 397-403).

Lines 401-419: It would be great to see how this in a figure – could you add a figure to the Supplement which compares the $\delta^{13}\text{C}$ and $\delta^{18}\text{O}$ of the different foram species and illustrates the corrections applied?
Done (Supplementary Figure S3-2).

Line 425: Please provide the mean value of the internal standard and external carbonate reference materials measured.
Done (Lines 493-498).

Line 501-503: Of what? Please specify.

We added the following sentences to the text (Lines 567-571):

“Wavelet spectral analysis was performed on the evenly spaced time series of the residue weight % (Figure 5), benthic $\delta^{18}\text{O}$ (Supplementary Figures S2-4c and S3-4a), orbital eccentricity (Supplementary Figure S3-4b) and XRF-scanner derived Log(Si/Ti) (Supplementary Figure S6-1). Raw data were sampled at 1 kyr resolution by linear integration using the resampling function in AnalySeries 2.08⁷⁵.”

Figure 1: I would suggest distinguishing the site U1490 from the other sites, by using different marker colour fill for example.
Done.

Figure 2: I am curious about the sharp decrease in CaCO₃ weight % at ~15.2 Ma. What is it, is this a known event?

Done. This is a volcanic tephra layer – now marked on the figure as T and explained in the figure caption (Line 916).as: T = volcanic tephra layer

Supplementary Information

Figure S1-1. Would be helpful to label the different water masses/currents in the figure.

Done.

Figure S1-2 and S1-2: These figures are very useful – does the distribution of $\delta^{13}\text{C}$, alkalinity and dissolved silicon look similar? Could you include them in figures as well?

Dissolved silica data exist for transects P11, P13 and P14. We added these transects in an additional supplementary figure (Supplementary Figure S1-3b, reproduced below).

Supplementary Figure S1-3b. Dissolved silica concentration along N-S meridional transects in the western Pacific at 155°E (upper panel left), 170°E (upper panel right) and 179°E (lower panel) from the WOCE Pacific Ocean Atlas4. Note that well-ventilated southern-sourced waters are silica- enriched.

Supplementary Information 2: Site U1490 revised composite depth scale and stratigraphic splice – I really appreciated this section. Would add to Fig. S2-1 mention of why Ba/Sr was used.

Done. We added this information to the text above the supplementary figure.

Table S2-2: Would suggest to also spell out again what r-mcd is in the caption.

Done.

Supplementary Data files – are overall very good and helpful. Please add foram species to “476349_0_data_set_8479709_s69p25”. Could you also provide the literature data from Figures S7-2 and S7-3 and Figure 5 in the main manuscript so that these figures can be easily redrawn by readers?

We added the species analysed for stable isotopes in Supplementary data file S3-2 and referred to the relevant data in the captions for Supplementary Figures S7-2 and S7-3 and Figure 5. We also added supplementary data files, which provide the data plotted in all main figures and Supplementary Figures S7-2 and S7-3. Data sources

files are also provided for these figures.

Reviewer #3 (Remarks to the Author):

The manuscript by Holbourn and co-authors presents a nice and well-written study. This is a great dataset and a sound interpretation of the data and for sure something that should be published in a journal like Nature Communications.

Still I have a couple of comments and suggestions that I hope make the manuscript better and in parts also better understandable for the reader.

We thank Reviewer 3 for detailed, critical comments that considerably helped us to improve the manuscript.

These are in the order they appear in the manuscript:

- Line 86: what is the estimated paleo water depth?

We estimated the paleo-water depth to be close to modern water depth, based on benthic foraminifer assemblage composition and comparison of benthic foraminifer $\delta^{18}\text{O}$ (Supplementary Table S7-1). We added a sentence to the main text to clarify the paleo-location and depth of the site, referring to Supplementary Table S7-1, which summarizes paleo-locations, paleo-water depths and relevant references for all sites referred to in this paper (Lines 102-104).

- Lines 144-148 and also line 175: Here I do not see why the authors argue for a larger ice sheet after Mi-2. For me the dataset in figure 2 suggests that there is a warming trend before MI-2 (at the onset of MCO) and the Mi-2 glaciation is just bringing the system back to 'normal' i.e. the values (and therefore in the interpretation of the $\delta^{18}\text{O}$ as an ice proxy) before the MCO warming.

We agree with the reviewer that the increase in mean $\delta^{18}\text{O}$ from 1.14‰ in the early part of the MCO (before Mi2) (16.9 to 16.1 Ma, n=159) to 1.34‰ after Mi2 (16.1 to 14.7 Ma, n=285) brings the system back towards background conditions prior to the onset of the MCO. We now make clear in the text that the mean $\delta^{18}\text{O}$ increase at ~16 Ma refers to the expansion of the reduced, unstable ice sheet that existed in the initial, warmer stage of the MCO (16.9-16.1 Ma) rather than a net increase in ice volume since 18 Ma. Please see revised text (Lines 169-172).

- Lines 170-174: "Between ~16.9 and 16 Ma, warmer intervals are characterized by increased concentration of biogenic silica and improved carbonate preservation at eccentricity/obliquity maxima. Conversely, silica accumulation decreased and carbonate dissolution increased during colder intervals at eccentricity/obliquity minima, as shown by increased abundance of etched or fragmented foraminifers."

This dissolution interval raises the question about preservation changes across the study interval. How good/bad are the foraminiferal shells preserved across the entire record and also across glacial/interglacial changes? This is something that should be discussed and shown to allow a judgement about the quality of the data.

To address this criticism:

(1) We added a plate with images of benthic foraminifers taken with a digital microscope to document the typical preservation state of species selected for isotope analysis during the MMCT and MCO (see Supplementary Figure S3-1 reproduced below). This plate includes examples of preservation across glacial-interglacial cycles between ~16.9 and 16 Ma (Figure S3-1-6 and -7).

(2) We added an overview of the preservation of planktic and benthic foraminifers within the whole interval studied in the Introduction (Lines 104-112) and in Supplementary Information 3. We also expanded descriptions of planktic assemblages during the MCO and MMCT (Lines 209-212, 225-227, 363-368).

Supplementary Figure S3-1: Digital light microscope images illustrating typical preservation state of late Early to Middle Miocene benthic foraminifer taxa selected for stable isotope analysis at Site U1490. 1. *Cibicidoides wuellerstorfi*, Sample U1490B-25H-5, 86-88 cm, umbilical view (13.70 Ma, 15.9% >63µm residue weight); 2. *Cibicidoides wuellerstorfi*, Sample U1490B-25H-5, 86-88 cm, umbilical view (13.70 Ma, 15.9% >63µm residue weight); 3. *Oridorsalis umbonatus*, Sample U1490A-27H-4, 92-94 cm, spiral view (16.93 Ma, 13.3% >63µm residue weight); 4. *Oridorsalis umbonatus*, Sample U1490A-27H-4, 92- 94 cm, umbilical view (16.93 Ma, 13.3% >63µm residue weight); 5. *Stilostomella subspinosa*, Sample U1490A-27H-2, 147-149 cm (16.61 Ma, 13.8% >63µm residue weight); 6. *Rectuvigerina* sp., Sample U1490A-27H-3, 6-8 cm (16.62 Ma, 5.9% >63µm residue weight); 7. *Cibicidoides mundulus*, Sample U1490B-25H-6, 96-98 cm, umbilical view (13.80 Ma, 5.7% >63µm residue weight); 8. *Cibicidoides mundulus*, Sample U1490B-25H-6, 96-98 cm, spiral view (13.80 Ma, 5.7% >63µm residue weight). Scale bars equal 100 µm. Note: *Stilostomella subspinosa* was not used for stable isotope analysis, but documents typical preservation state.

- Line 229: There is no d18O shown in figure 5

"By contrast, Site 1146 (paleolatitude: ~25°N, paleodepth: ~2000 m) displays consistently lower d13C and d18O throughout the MCO (Figure 5)"

Thank you for pointing this out. We have now revised the text, adding a reference to Supplementary Figures S7-2 and S7-3, which show comparisons of both $\delta^{18}\text{O}$ and $\delta^{13}\text{C}$ at Sites 1146 and U1490 (Lines 262-263).

- Chapter starting at line 212: This chapter is internally consistent in its interpretation of the data. But I miss a clear and convincing statement at the beginning that all of these data reflect SCW and not another water mass like North Pacific waters which might still be different from PIW at Site 1146 (see e.g. studies of Burls et al). As a reader this chapter seems to build on the assumption that Site U1490 is bathed by SCW but without explanation or justification.

In answer to this criticism, we revised the introductory paragraph of this chapter (Lines 246-249). The assumption that Site U1490 is bathed by SCW is primarily based on the similarity between the $\delta^{13}\text{C}$ records from Southern Ocean Sites 751 and 1171 (paleolatitude: ~55-56°S, paleodepth: ~1500 m) and western equatorial Pacific Site U1490 throughout the MCO.

- Line 274-275: sorry, but I do not see the proposed relation between carbonate-depleted and colder periods in the interval between 16.9 and 16.2 Ma in figure 3.

"The preferential occurrence of carbonate-depleted intervals during colder periods between ~16.9 and 16 Ma (Figure 3) suggests that the formation of a carbonate undersaturated water mass intensified, when Antarctica retained a sizeable ice cover and sea 276 surface temperatures decreased."

To highlight the relationship between carbonate-depletion and colder periods and the overall glacial-interglacial variability of the carbonate proxy records from ~16.9 to 16 Ma, we shaded warmer periods in light orange in Figure 3 (see revised figure below).

Figure 3: Expanded view of interval 17-15.2 Ma at WPWP Site U1490. **a** Benthic foraminifer $\delta^{13}\text{C}$. Monterey Excursion carbon isotope maxima^{82,94} are labeled CM1, 2, 3, 4a. **b** Weight percentage of >63 μm coarse fraction residues (exclusively composed of complete or fragmented carbonate foraminifer tests) as indicator for carbonate dissolution. **c** Wavelet power of >63 μm coarse fraction residues. Note dominance of 41 kyr (obliquity) and 100 kyr (eccentricity) periodicities (indicated by dashed black lines) and change in amplitude variability across Mi2 transition. **d** Raw and 9 point moving average XRF scanner-derived Log(Si/Ti) as indicator of biogenic silicate (opal). **e** Benthic foraminifer $\delta^{18}\text{O}$. Note $\delta^{18}\text{O}$ maximum (indicated by blue arrow) and ~0.2‰ increase in $\delta^{18}\text{O}$ mean following Mi2 event (shaded light blue). **f** Eccentricity and eccentricity-obliquity (tilt) composite (ET) from La04⁴³.

Revised Figure 3

- Line 290ff: but CaCO_3 is decreasing well before Mi-2! So how reliable is the stated speculation?

“We speculate that the larger, more stable ice sheet induced surface cooling close to Antarctica after ~16 Ma, promoting increased formation and northward expansion of corrosive deep waters. Increased ocean acidification despite enhanced carbonate productivity in the mid and low latitudes³⁹⁻⁴¹ would have bolstered formation of carbonate undersaturated deep water in a high $p\text{CO}_2$ Miocene world.”

We agree that carbonate dissolution started after ~16.3 Ma and that a first dissolution maximum occurred at ~16.2 Ma. However, carbonate preservation temporarily recovered at ~16.1 Ma during an eccentricity maximum, reflected by a sharp minimum in benthic $\delta^{18}\text{O}$ (Figure 3, panels b and e). We therefore consider this transient

increase in dissolution to represent a precursor event, whereas the main change point in carbonate dissolution occurred at the end of Mi2 (~16.0 Ma), coincidentally with the beginning of an increase in mean $\delta^{18}\text{O}$ by ~0.2 ‰ (Figure 3, panels b and e). Carbonate dissolution reached a maximum after Mi2 (~16.0 Ma) and remained high over the following 600-700 kyr (Figure 3, panel b).

- Line 319: where is the entire Log Si/Ti record shown? Would be good to have this in one of the main figures.

We followed the reviewer's suggestion and added the entire Log Si/Ti record to Figure 7.

- Figure 5: why is Site 1146 in panel d not plotted against the Site U1490 record as all the other sites?

In answer to this criticism, we have now added (1) a comparison of the Sites 1146 and U1490 $\delta^{13}\text{C}$ records to Figure 5 (Figure 5Ad) as well as (2) a second panel to Figure 5 (Figure 5B), which more clearly highlights the temporal evolution of $\delta^{13}\text{C}$ gradients between Site U1490 and the other sites (including Site 1146) referred to in this paper. Please see also response to Reviewer 1 above (p. 3), which includes the new Figure 5B.

REVIEWERS' COMMENTS

Reviewer #1 (Remarks to the Author):

Dear authors,

Thank you very much for addressing all comments from my side.

The study site IODP Site U1490 recorded the deep water inflow from the Southern Ocean along the western Pacific margin. It provides a crucial record of the deeper limb of the Pacific overturning circulation. Moreover, the southern-sourced waters exert a major control over the entire Pacific overturning circulation as these waters serve as a reservoir of heat energy and carbon.

Given the value of this continuous high-resolution stable isotope data from the Pacific Ocean, and also their effort and satisfactory results in revising the manuscript, I would like to recommend to accept the manuscript in its current form.

Reviewer #2 (Remarks to the Author):

Review of revised manuscript NCOMMS-476349 "Re-organization of Pacific overturning circulation across the Miocene Climate Optimum" by Ann Holbourn, Wolfgang Kuhnt, Denise K. Kulhanek, Greg Mountain, Yair Rosenthal, Takuya Sagawa, Julia Lübbbers, Nils Andersen

I would like to acknowledge the authors' careful revisions and thoughtful responses. I particularly appreciated the addition of the section "Perspectives" – I think this makes a wonderful addition that provides further context and will help steer future research efforts. It is a shame that a modelling component could not have been incorporated; that said I understand the challenges the authors mention and thus consider this point sufficiently addressed.

I have few final, minor (mostly editorial) suggestions for the authors:

Line 57: "irreversible" – would not use this wording, if anything, the geological record shows us that climate can change from greenhouse to icehouse states back and forth and thus is reversible (the right question perhaps is on what time scales). Indeed, in line 71 you mention that the MCO "reversed" the long-term cooling trend, arguing that climate trends can be reversed.

Line 80: I am missing here a one final sentence outlining the "knowledge gap"– despite the previous work you mention, what is still unclear (that you then address?)?

Lines 105-106: I would avoid unnecessary abbreviations. APC and HAPC are only used in line 110, could easily remove these.

Lines: 118-121: Please add references to climate trends, ice volume, greenhouse gas concentrations.

Line 134: after "ODP Site 806", in brackets for example, would be helpful to add relative position to your site U1490. It would also help to indicate the position of this site in Figure 1.

Line 190: "Our results" – please specify what results/data you refer to (for non-specialist audience)

Line 231-232: "major atmospheric pCO₂ and ocean acidification increases in Earth's history" Sounds a bit strange, would rephrase, maybe: "major intervals of atmospheric CO₂ increase and ocean acidification"?

Line 403-423: Nice, I really appreciate this addition!

All the best to your research!

Reviewer #3 (Remarks to the Author):

After carefully reading the reply letter and the new version of the manuscript, I have to admit that the authors have done a great job in taking into account the comments and corrections by the three reviewer. Although I personally would have hoped for some more details in a few instances, this is a matter of personal preferences and I fully understand why the authors have responded and changed the text in the way they have done it.

Given the good job and the overall well-developed revised version of the manuscript I have no further comments and suggestions and fully support the publication of this very nice record.

ORIGINAL MANUSCRIPT NCOMMS-23-63602A

REVIEWERS' COMMENTS

Reviewer #1 (Remarks to the Author):

Dear authors,

Thank you very much for addressing all comments from my side.

The study site IODP Site U1490 recorded the deep water inflow from the Southern Ocean along the western Pacific margin. It provides a crucial record of the deeper limb of the Pacific overturning circulation. Moreover, the southern-sourced waters exert a major control over the entire Pacific overturning circulation as these waters serve as a reservoir of heat energy and carbon.

Given the value of this continuous high-resolution stable isotope data from the Pacific Ocean, and also their effort and satisfactory results in revising the manuscript, I would like to recommend to accept the manuscript in its current form.

Thank you again for your constructive review.

Reviewer #2 (Remarks to the Author):

Review of revised manuscript NCOMMS-476349 "Re-organization of Pacific overturning circulation across the Miocene Climate Optimum" by Ann Holbourn, Wolfgang Kuhnt, Denise K. Kulhanek, Greg Mountain, Yair Rosenthal, Takuya Sagawa, Julia Lübbers, Nils Andersen

I would like to acknowledge the authors' careful revisions and thoughtful responses. I particularly appreciated the addition of the section "Perspectives" – I think this makes a wonderful addition that provides further context and will help steer future research efforts. It is a shame that a modelling component could not have been incorporated; that said I understand the challenges the authors mention and thus consider this point sufficiently addressed.

I have few final, minor (mostly editorial) suggestions for the authors:

RESPONSE TO ADDITIONAL COMMENTS FROM REVIEWER 2

Line 57: "irreversible" – would not use this wording, if anything, the geological record shows us that climate can change from greenhouse to icehouse states back and forth and thus is reversible (the right question perhaps is on what time scales). Indeed, in line 71 you mention that the MCO "reversed" the long-term cooling trend, arguing that climate trends can be reversed.

Thank you for this helpful suggestion. We replaced "irreversible" by "abrupt" (Line 56).

Line 80: I am missing here a one final sentence outlining the "knowledge gap" – despite the previous work you mention, what is still unclear (that you then address)?

The following sentence was added on Lines 79-82: "However, the evolution of the overturning circulation and deep water chemistry across these major climate reversals remain poorly understood due to the scarcity of continuous, well dated sedimentary records spanning the MCO and MMCT."

Lines 105-106: I would avoid unnecessary abbreviations. APC and HAPC are only used in line 110, could easily remove these.

We decided to retain these two abbreviations because they are used again a few lines below (Line 112) and also in the "Methods" section (Lines 453-454).

Lines: 118-121: Please add references to climate trends, ice volume, greenhouse gas concentrations.

References added (Line 123).

Line 134: after "ODP Site 806", in brackets for example, would be helpful to add relative position to your site U1490. It would also help to indicate the position of this site in Figure 1.

We added the position of Site 806 relative to that of Site U1490 on Lines 136-137. We also indicated the position of Site 806 on Figure 1.

Line 190: "Our results" – please specify what results/data you refer to (for non-specialist audience)

We changed the text to: "Our high-latitude climate and deep water chemistry proxy records...." (Lines 193-194).

Line 231-232: "major atmospheric pCO₂ and ocean acidification increases in Earth's history" Sounds a bit strange, would rephrase, maybe: "major intervals of atmospheric CO₂ increase and ocean acidification"?

Thank you for pointing this out. We changed the text to “intervals of major atmospheric CO₂ increase and ocean acidification” (Lines 235-236).

Line 403-423: Nice, I really appreciate this addition!
Many thanks for your constructive comments.

All the best to your research! Thank you!

Reviewer #3 (Remarks to the Author):

After carefully reading the reply letter and the new version of the manuscript, I have to admit that the authors have done a great job in taking into account the comments and corrections by the three reviewer. Although I personally would have hoped for some more details in a few instances, this is a matter of personal preferences and I fully understand why the authors have responded and changed the text in the way they have done it.

Given the good job and the overall well-developed revised version of the manuscript I have no further comments and suggestions and fully support the publication of this very nice record.

Thank you again for your constructive review.